# Role of stem-like cells in chemotherapy resistance and relapse in pediatric T-cell acute lymphoblastic leukemia

Julia Costea [1,2,3,17], Kerstin K. Rauwolf[4,17], Pietro Zafferani [2,5], Tobias Rausch [1,2,6], Anna Mathioudaki [1,7,8], Judith Zaugg [1,7], Martin Schrappe[9], Cornelia Eckert [10,11], Gabriele Escherich[12], Jean P. Bourquin [4], Beat Bornhauser [4], Andreas E. Kulozik [1,13,14,15] ✉ & Jan O. Korbel [1,2,16] ✉

T-ALL relapses are characterized by chemotherapy resistance, cellular diversity and dismal outcome. To gain a deeper understanding of the mechanisms underlying relapses, we conduct single-cell RNA sequencing on 13 matched pediatric T-ALL patient-derived samples at diagnosis and relapse, along with samples derived from 5 non-relapsing patients collected at diagnosis. This comprehensive longitudinal single-cell study in T-ALL reveals significant transcriptomic diversity. Notably, 11 out of 18 samples exhibit a subpopulation of T-ALL cells with stem-like features characterized by a common set of active regulons, expression patterns and splice isoforms. This subpopulation, accounting for a small proportion of leukemia cells at diagnosis, expands substantially at relapse, indicating resistance to therapy. Strikingly, increased stemness at diagnosis is associated with higher risk of treatment induction failure. Chemotherapy resistance is validated through in-vitro and in-vivo drug testing. Thus, we report the discovery of treatment-resistant stem-like cells in T-ALL, underscoring the potential for devising future therapeutic strategies targeting stemness-related pathways.

ALL is the prevalent form of pediatric leukemia affecting approximately 1 out of 3 of pediatric cancer patients. T-ALL comprises ~15% of pediatric ALL and can be cured in ~80% of affected patients[1]. However, the outcome in relapsing patients is dismal with a prognosis of merely 20% survival[2].

T-ALL arises in the thymus from abnormal T-cell development. T-ALL subtypes were initially classified based on their resemblance to thymocytes at different stages of maturation[3]. Advances in sequencing have since revealed additional molecular diversity within T-ALL, allowing for subtype classification based on the coordinated dysregulation of gene expression patterns caused by the activation of distinct transcription factors, such as TAL1, LMO1/2, TLX1/3, and NKX2 and others[4]. Although the acquisition of mutations and changes in the expression of genes with roles in pharmaco-resistance,

such as NOTCH1, JAK-STAT, PI3K-AKT or RAS-MAPK pathway genes, as well as in epigenetic regulation, such as members of the PRC2 complex or CREBBP, have previously been described[5–11], a unifying mechanism explaining the development of relapse remains unknown.

A differential analysis of disease progression and relapse in T-ALL has been facilitated by the identification of two types of relapses in T-ALL: type-1, originating from the major clone of the initial disease and type-2, originating from a minor ancestral clone[5,7]. Type-1 relapses are characterized by frequent activation of the IL7 receptor pathway, whereas type-2 relapses are driven by activation of the transcription factor TAL1 and often harbor mutations in genes previously linked to germline cancer predisposition[7,12]. Considering that a substantial proportion of relapses originate from a minor diagnostic subclone, bulk

genetic and transcriptomic analyses are insufficient for characterizing treatment resistance acquisition.

To comprehensively investigate the heterogeneity of T-ALL in single cells and gain a better understanding of mechanisms driving T-ALL relapse, we therefore conduct single-cell full-length total RNA sequencing on patient-derived samples with VASA-seq[13], with the aim to follow the evolution of treatment-resistant cell populations from initial diagnosis to relapse in the same patient. For our study, we utilize cells from patient-derived xenografts (PDXs), ensuring sufficient availability of sample material. Our analysis in PDX models of 18 T-ALL patients identifies a cell population that converges at a gene-regulatory network revealing a common dormant stem-like cell phenotype with resistance to chemotherapy in functional assays.

**Fig. 1 | Expansion of a stem cell-like immature dormant subclone during the development of relapse in the PDX-derived cells of a child with T-ALL. a** Scheme of the experimental procedure. Created in BioRender. Costea, J. (2025) https://BioRender.com/8qu9bku. The samples have been obtained from patient P2 at the time of initial diagnosis (Ini) and relapse (Rel) and were engrafted in immunodeficient mice. T-ALL cells have been extracted from the PDX mice and analyzed by VASA-seq. **b, d, e:** UMAP illustration of the cellular composition at initial diagnosis ($n = 653$ cells from one sample) (**d**) and relapse ($n = 578$ cells from one sample) (**e**) and combined ($n = 1231$ cells from two samples) (**b**) after graph-based clustering analysis using the Louvain algorithm. **c** stacked barplot illustrates the cellular composition at initial diagnosis and relapse. **f, g:** distribution of cells ($n = 1231$ cells from two samples) in different cell cycling phases as inferred by the transcriptional profile visualized by UMAP (**f**) and stacked barplot (**g**). **h, i:** distribution of predicted cell types after mapping cells ($n = 1231$ cells from two samples) onto a human thymic reference dataset[16] visualized by UMAP (**h**) and stacked barplot (**i**). **j** Average enrichment and depletion of regulons ($n = 11$) in cluster 2 identified by pySCENIC (Fisher's exact test, log2FC > 0.25, padj < 0.05). **k** Gene set enrichment analysis (GSEA) plot of GO biological processes shows significant enrichments of cluster 2 (red) vs rest (blue) ($n = 1231$ cells from two samples) based on 1288 differentially expressed genes (Non-parametric Wilcoxon test, log2FC > 0.5 and padj < 0.05). **l** Illustration of stem-like cell features vs features of the two major cell clusters. Created in BioRender. Costea, J. (2025) https://BioRender.com/zqm1nya. Source data are provided as a Source Data file.

## Results

### Expanding stem-like cells revealed from a PDX model of a relapsing T-ALL patient

To investigate differences in the individual cellular architecture of T-ALL during disease progression, we first conducted VASA-seq on initial and relapse-derived samples of our index patient (P2) (Fig. 1, Supplementary Fig. 1, and Supplementary Data 1), generating data for 1231 single cell full-length transcriptomes altogether (initial: 653; relapse: 578). Graph-based clustering analysis using the Louvain algorithm identified a predominant cell population (cluster 0, 98.62% of the total population) at initial diagnosis, which decreases to 3.27% in relapse. Conversely, an initial minor cell population (cluster 2) expands from 1.37% (6 cells) at diagnosis to 26.47% (153 cells) at relapse suggesting treatment resistance and clonal selection. In addition, this sample acquired a second relapse-specific cell population that has not been detected at initial disease (cluster 1, 70.24%) (Fig. 1b–e).

Subsequent assessment of these clusters in terms of cell cycle and developmental stage (Supplementary Methods) revealed a significantly higher fraction of cells in the G1 phase of the cell cycle for cluster 2 (88.27% in G1) compared to the other clusters (permutation test: FDR < 0.05, log2FD > 0.25) (Fig. 1f–g and Supplementary Fig. 1a), suggesting a lower cell cycling activity in this cell population. This finding is further supported by the decreased activity of cell cycling TFs (TFDP1, E2F1), which we identified when applying the pySCENIC tool for single-cell regulatory network inference and clustering (pySCENIC) (Fig. 1j and Supplementary Data 2) (Supplementary Methods)[14,15]. When mapped onto a thymic single cell reference[16], cluster 2 significantly differs from cluster 0 and cluster 1 (permutation test: FDR < 0.05, log2FD > 0.25) (Fig. 1h, i and Supplementary Fig. 1b) by almost completely resembling a more immature double negative (DN) T-cell progenitor phenotype (96.30% DN, 3.70% DP). In contrast, the transcriptional profiles of the other clusters resemble a mixture of the DN and the double positive (DP) T-cell progenitor phenotype (cluster 0: 47.96% DN, 52.04% DP, cluster 1: 71.18% DN, 28.81% DP). To investigate a potential bone marrow progenitor origin of the more immature cell population, we mapped our T-ALL dataset against a bone marrow reference dataset (Supplementary Fig. 1d–f)[17]. However, this analysis did not indicate that these cells derive from an earlier hematopoietic stage beyond the lymphoid lineage. Instead, cluster 2 predominantly resembled CD4 memory cells, a population characterized by pronounced quiescence[18], while the majority of cells in cluster 0 and cluster 1 aligned most closely with common lymphoid progenitors.

In a next step, we investigated functional differences of these dormant, immature cells of cluster 2 compared to leukemic cells of the other clusters utilizing gene set enrichment analysis in Python (i.e., pyGSEA; Supplementary Methods) (Fig. 1k)[19–21]. A significant enrichment of cell migration (FDR < 0.05, NES 2.07) and plasma membrane organization (FDR < 0.1, NES 2.02) alongside a reduction of cell cycling (FDR < 0.05, NES −2.40) and metabolic processes (FDR < 0.05, NES −1.98) as well as nucleosome organization (FDR < 0.05, NES −2.31) resemble characteristics which have been previously noted in stem-like cells[22,23] (Fig. 1l).

Taken together, VASA-seq analysis in PDX-derived cells of this index patient revealed a small cell population at initial diagnosis of cells exhibiting dormant stem cell-like properties, which is substantially expanded during relapse.

### pySCENIC identifies a uniform population among TAL1 driven T-ALLs

We next expanded our analysis to a group of 9 additional TAL1 driven T-ALLs which either did not (PDX-derived samples of 5 patients) or did (PDXs of 4 patients) develop a type-2 relapse following first-line treatment thus increasing the total number of patients analyzed to 10 (Fig. 2a). We performed VASA-seq, generating data for 7929 additional (total: 9160) single-cell full-length transcriptomes (Fig. 2 and Supplementary Fig. 2). To analyze each single cell with respect to transcriptomic state and regulatory networks, we performed graph-based clustering based on regulon activities inferred from pySCENIC[14,15]. We identified cells enriched in both recently distinguished TAL1 subtypes —the αβ-like and DP-like TAL1 T-ALL (Supplementary Fig. 2d, e)[24]— successfully separating both subtypes based on their distinct regulon activity profiles. The DP-like subtype included PDXs of relapsing patients P12, P10, P6, and P2, and non-relapsing patient P68, while the αβ-like subtype included PDXs of relapsing patients P6 and P8 and non-relapsing patients P41, P52, P59, and P63. Interestingly, we observed P6 PDX cells reflecting both subtypes.

In addition, while the majority of cells clustered by individual patient (Fig. 2b), a small proportion of cells from different PDXs representing both TAL1 subtypes was grouping closely, indicative for a cell subpopulation driven by a set of common regulons (Fig. 2c).

As previously seen in cluster 2 of our index sample, a predominant fraction of cells in this shared subpopulation showed arrest in the G1 phase (75.75%), with the remaining 24.25% progressing through S/G2M phases. This is in stark contrast to the heterogenous predominant subpopulations of the individual patients' PDXs, which displayed significantly higher rates of cell division (28.90% G1, 71.09% in S/G2M, permutation test: FDR < 0.05, log2FD > 0.25) (Fig. 2d, e and Supplementary Fig. 2c). These data imply the existence of a small and distinct cell population shared across different T-ALL-derived samples, marked by low cell division, which we subsequently characterized in further detail.

Notably, when assessing these single cells in terms of their developmental stage, it became apparent that the transcription profile of this quiescent cell population predominantly resembles immature double negative (DN) T progenitor cells (66.90%, 10.25% DP). By contrast, the highly proliferative population mostly comprises cells with a transcriptional profile resembling double positives (DP) (40.99% DN, 56.98% DP, permutation test: FDR < 0.05, log2FD > 0.25) (Fig. 2g, h and Supplementary Fig. 2f).

We delved deeper into the single cell transcriptomic data to investigate specific pathways active in this dormant cell population.

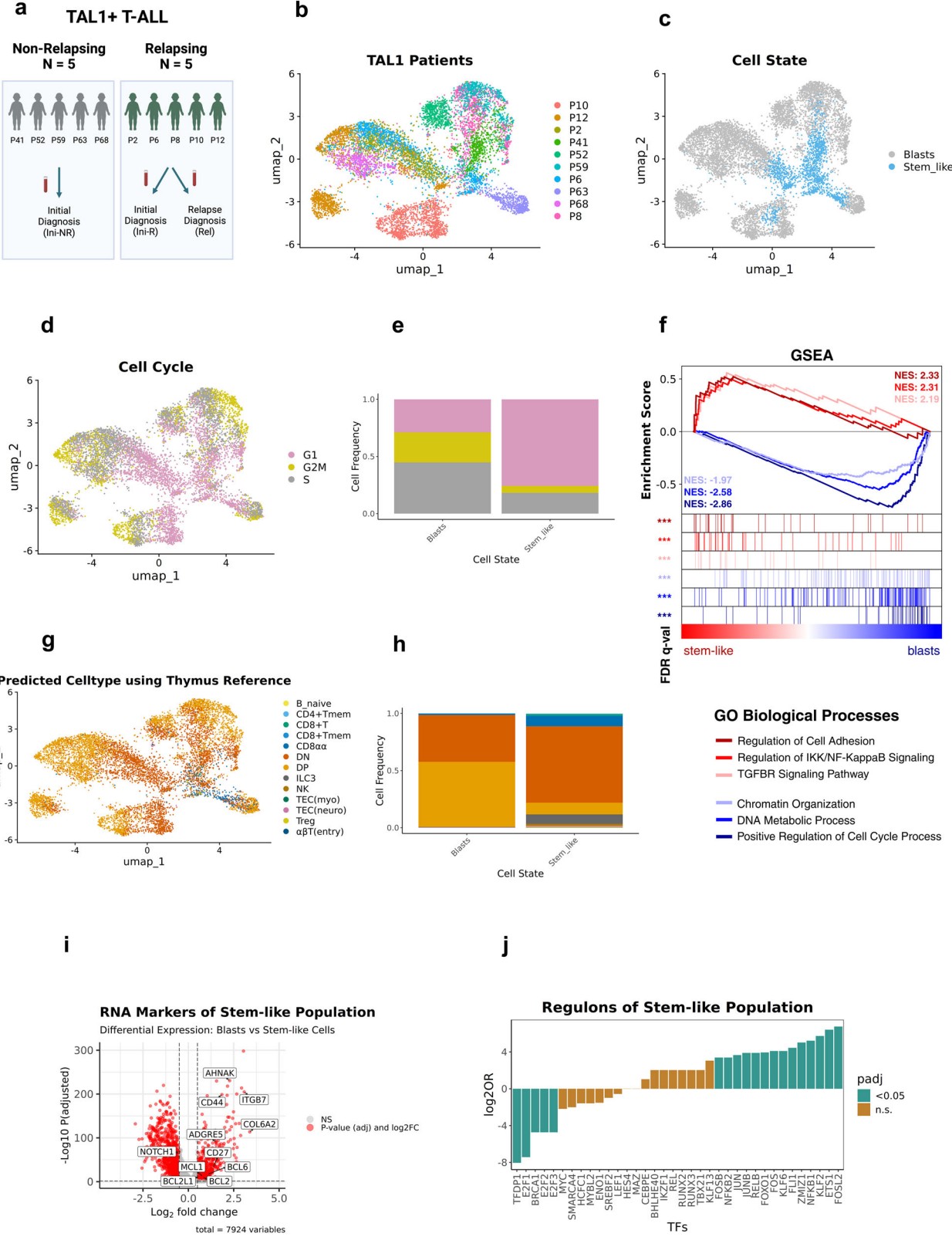

Performing GSEA revealed a strong enrichment in cell adhesion (NES 2.33, FDR < 0.05), as well as NF-κB (NES 2.31, FDR < 0.05) and TGF-β signaling (NES 2.19, FDR < 0.05) which are known to enforce drug resistance[25,26] (Fig. 2f). Additionally, unlike the cell population with a transcriptional profile of rapidly proliferating cells, the cells with a dormancy phenotype showed a marked downregulation of metabolic processes (NES −2.86, FDR < 0.05, FDR), a common characteristic of stem-like cells[23]. In addition, we find significant enrichment of pathways linked to T-cell quiescence (NES 2.07, FDR < 0.01)[27] as well as clinical correlates of treatment resistance such as prednisone resistance (NES 1.94, FDR < 0.01)[28] and persistent minimal residual disease (NES 1.78, FDR < 0.05) (MRD)[29] (Supplementary Fig. 2j–l) in the dormant population. We reasoned that presence of a common cell subpopulation characterized by stem-cell like properties could be of

**Fig. 2 | pySCENIC analysis reveals a shared regulon activity in small subclones in TAL1-driven T-ALL PDXs with stem cell-like properties. a** Illustration of a TAL1-driven T-ALL patient cohort consisting of 5 non-relapsing patients (Ini-NR, grey), and 5 type-2 relapsing patients (dark green). Samples have been taken during initial diagnosis (Ini-R) and relapse (Rel) and engrafted in mice. Created in BioRender. Costea, J. (2025) https://BioRender.com/hwpx34k. **b–d, g** Analysis based on pySCENIC regulon activity visualized by UMAP ($n = 9160$ cells from 15 samples, 10 patients). **b** distribution of TAL1-driven T-ALL PDXs. **c** assignment of stem-like cells and highly proliferative cell population in TAL1 PDX-based cohort. **d** distribution of cells in different cell cycling phases inferred by transcriptional profile. **g** distribution of predicted cell types after mapping cells onto a human thymic reference dataset[16]. **e, h** Stacked barplots comparing distribution in stem-like cells and other leukemic blasts. **e** distribution of predicted cell cycling phases.

**h** distribution of predicted cell types. **f** Gene set enrichment analysis (GSEA) plot of GO biological processes shows significant enrichments of pathways in stem-like cells (red) vs other leukemic blasts (blue) ($n = 9169$ cells from 15 samples, 10 patients) based on differential expression of 3296 genes (Non-parametric Wilcoxon test, log2FC > 0.5 and padj < 0.05). **i** Volcano plot displaying enrichment and depletion of RNA markers ($n = 1705$) in stem-like T-ALL PDX-derived cells ($n = 9169$ cells from 15 samples, 10 patients). red: differentially expressed markers (Non-parametric Wilcoxon test, log2FC > 0.5 and padj <0.05). grey: non-significant changes of expression. Annotated markers are further described in the results section. **j** Average enrichment and depletion of regulons ($n = 37$) in stem-like cells of all TAL1 samples ($n = 9169$ cells from 15 samples, 10 patients) (Fisher's exact test, log2FC > 0.25, padj < 0.05 highlighted in green, non-significant displayed in brown). Source data are provided as a Source Data file.

potential relevance for T-ALL therapy resistance, since chemotherapy preferentially targets dividing cells.

## RNA markers and active regulons of the stem-like cell population

To gain further insights into the transcriptional program of these cells, we next performed comprehensive differential expression analyses (Fig. 2i and Supplementary Data 3). We found that the stem-like cells, unlike the other cell subpopulations, are characterized by a similar RNA expression pattern, including several genes that have been recently reported to play a role in stem-like cells in a murine T-ALL model, e.g., CD44[30], RORA, CD226, CD52[22]. Notably, we observed markers that have been suggested to play a role in treatment resistance but were not described in the concept of stemness in T-ALL, e.g., ADGRE5 (CD97)[31], AHNAK[32], CD27[33], COL6A2[34], ITGB7[31], several members of the S100 family[35] and the anti-apoptotic proteins BCL-6, BCL-2, BCL2L1/BCL-xL, and MCL1. Remarkably, NOTCH1 is downregulated in the stem-like cell population.

We next utilized pySCENIC, allowing us to identify those TFs that drive the expression of most of the genes in the cluster and therefore are likely to play a particularly relevant biological role (Fig. 2j and Supplementary Data 4). We identified five TFs that were significantly less active in the stem-like cells (Fisher's exact test, padj < 0.05), all of which are involved in cell cycling (TFDP1, E2F1, BRCA1, E2F2, E2F3), in further support of the quiescent phenotype of these cells. The 14 TFs with significantly enriched activity in the stem-like cell population (Fisher's exact test, padj < 0.05) are grouped into six different TF families—KLF, AP1, NF-κB, ETS, FOXO, and PIAS family—all of which are key regulators of hematopoietic development. Notably, KLF2, one of the significantly enriched TFs, represents a main driver of quiescence, migration, and anti-apoptotic signaling in T cells[36–38].

## Proportion of stem-like cells in T-ALL subgroups and relapse types

We next investigated whether these stem-like cells are unique to the TAL1 subgroup or are also present in the NKX2, HOXA, and TLX1/3 subgroups of T-ALL. Considering that TAL1-driven T-ALLs most frequently relapse as type-2 and the other subgroups frequently relapse as type-1[5,7], we also considered the relapse-type in our analysis. We thus expanded our PDX-based cohort, now encompassing VASA-seq samples from 13 relapsing (5 type-1, 8 type-2) and five non-relapsing patients of whom 11 were TAL1-, three TLX1/3-, two NKX2-, and two HOXA-driven. While 10 out of the 11 TAL1 driven patients showed a type-2 relapse only one experienced a type-1 relapse (P11). Of the 7 patients with the less common drivers four experienced a type-1 and three a type-2 relapse (Fig. 3a).

VASA-seq applied to these PDX samples resulted in 9718 additional (total: 18,878) sequenced full-length transcriptomes (Fig. 3 and

Supplementary Fig. 3). Utilizing the pySCENIC output to perform dimensional reduction on all samples together revealed a higher degree of heterogeneity among individual PDX samples of the other T-ALL subgroups compared to the TAL1 subgroup (Fig. 3b, c). To identify stem-like cells in the dimensional space, we used the set of 601 differentially expressed marker genes (log2FC > 0.5, padj < 0.05) of the previously defined stem-like cell population of the TAL1 samples (Fig. 2i and Supplementary Data 3) to create a T-ALL 'stemness' signature (Supplementary Methods), and scored each cell according to their relative expression of stemness genes (i.e., the stemness score) (Fig. 3d, e and Supplementary Methods). We identified stem-like cells to have a frequency of 13.53% in the total population of cells (2554 of 18,878 cells), with variations seen among individual patients, subgroups, and relapse types. Within the TAL1 subgroup, these data further corroborate the observation that the stem-like cells are more homogenous than the major leukemic cell population. As a notable exception, P11, the only TAL1 sample with a type-1 relapse, neither clustered with other TAL1 samples nor showed an enrichment in stemness. We did, however, observe enrichment in the stemness score in the other T-ALL subgroups suggesting that stem-like cells are a common phenomenon in T-ALL and not exclusive for TAL1 patients. Nonetheless, the proportion of stem-like cells significantly differs among the subgroups and the type of relapse, being most frequent in those samples representing TAL1 patients (18.86% of TAL1 T-ALL cells) and type-2 relapses (15.41% of type-2 relapse T-ALL cells) (permutation test: FDR < 0.05, log2FD > 0.25) (Fig. 3f, g and Supplementary Fig. 3a, b), and less common in PDXs of TLX1 patients (2.56% of TLX1-driven T-ALL cells) and type-1 relapses (3.70% of type-1 relapse T-ALL cells). We also deconvoluted the stemness score in PDX-derived cells of individual patients (Fig. 3h), categorizing samples based on the proportion of stem-like cells into one of three categories: high (>15%), moderate (5–15%), and low (<5%). 8 of 11 TAL1 PDX samples contained a high or moderate stem-like cell population. Of those, 3 of 5 represented samples from non-relapsing patients and 5 of 6 samples from relapsing patients, suggesting that the presence of a stem-like cell population does not per se predict the risk of relapse. Further, 1 of 2 NKX2 PDX samples and 1 of 2 HOXA PDX samples contained a high frequency of stem-like cells. In contrast, none of the TLX1 PDXs contained a high frequency and only 1 of 3 showed a moderate frequency of stem-like cells.

## Common patterns of alternative splicing enriched in stem-like cells

Increasing evidence has emerged that alternative splicing (AS) is highly correlated with ALL treatment resistance[39–41], and acts as a potential driver in pediatric cancers[42]. Harnessing the full-length transcriptome dataset provided through VASA-seq, we investigated splicing usage differences between stem-like cells and the remaining T-ALL cell populations in our PDX cohort. We employed a

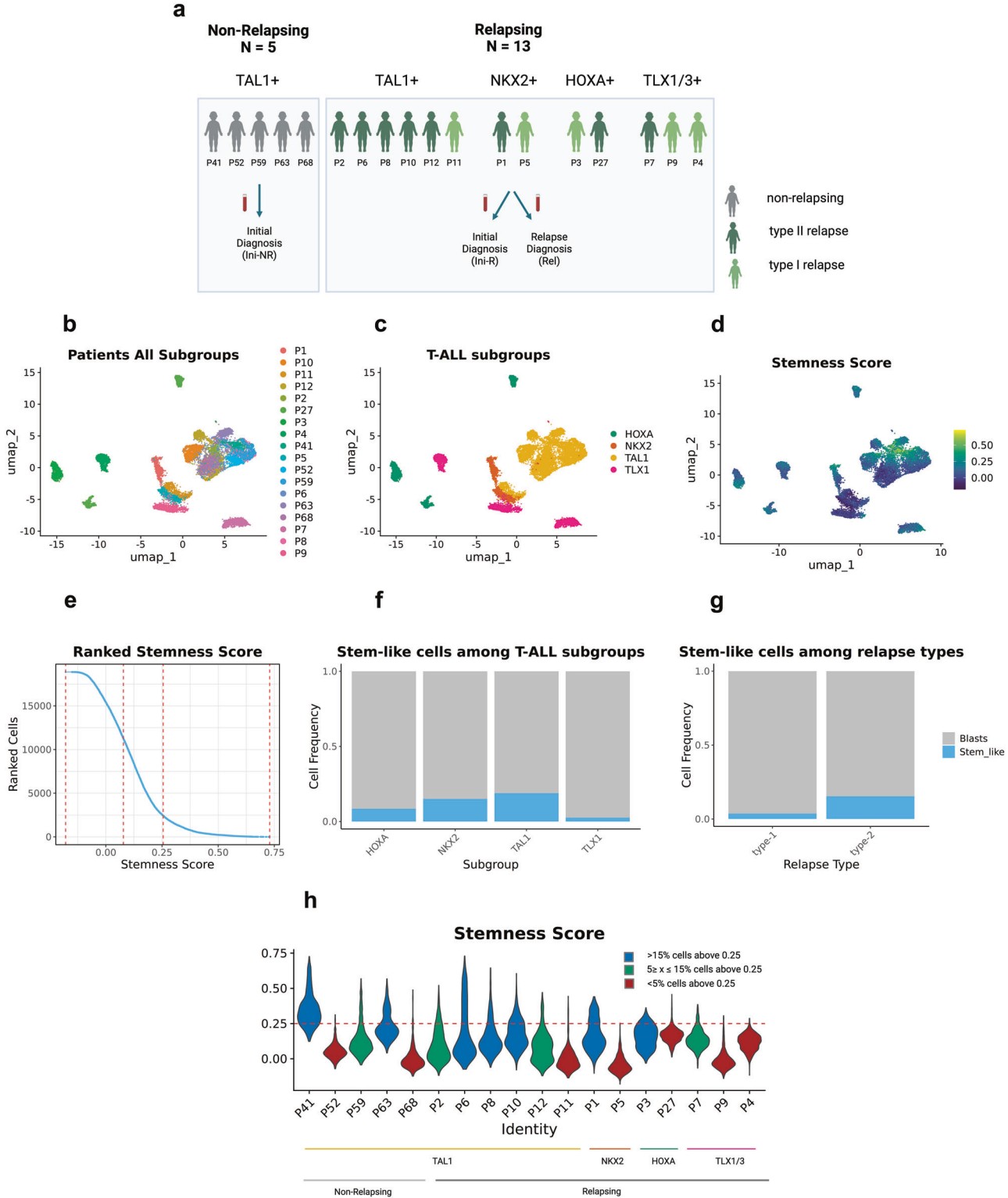

previously established computational workflow[13] to identify significant changes of AS patterns in the stem-like cell population (Supplementary Methods, Supplementary Fig. 4a, b). Analysis of the different splicing event types revealed the prevalence of four AS event types enriched in the stem-like cell population: intron retention, core exon skipping, usage of an alternative donor, and use of an alternative acceptor site (Fig. 4a, b). The number of total AS events detected in each sample varies from patient to patient (28-356), but is not linked to the number of cells sequenced per PDX or

the size of the stem-like cell population (Supplementary Data 6 and Supplementary Fig. 4c, d).

We next elaborated on whether a subset of these genes affected by AS show recurrence across several T-ALL PDXs, (Fig. 4c), implying they could potentially serve as a future treatment target. We find 161 genes that are affected by AS in samples of at least two patients, including known T-ALL oncogenes (e.g., LCK, MYB, FOS; detected in four patients) and tumor suppressors (e.g., PTPRC; detected in four patients), splicing factors (e.g., HNRNPH1; detected in six patients), ribosomal proteins

**Fig. 3 | Stem-like cells are more prevalent in PDXs of TAL1-driven T-ALL than in the other subgroups. a** Illustration of patient cohort consisting of 5 non-relapsing patients (grey), 8 type-2 relapsing patients (dark green), and 5 type-1 relapsing patients (light green). Samples reflect four different T-ALL subgroups: TAL1 (11×), NKX2 (2×), HOXA(2×), TLX1/3 (3×). Samples have been obtained either at initial diagnosis (Ini-NR and Ini-R) or at relapse (Rel) and engrafted in mice. Created in BioRender. Costea, J. (2025) https://BioRender.com/pe6xxq3. **b**–**d** Graph-based clustering analysis including PDX-derived cells of 18 patients based on pySCENIC regulon activity and visualized by UMAP ($n = 18,878$ cells from 31 samples, 18 patients). **b** distribution of PDX-derived cells from individual patients. **c** distribution of T-ALL subgroups. **d** enrichment of stemness score: Score is based on the average expression of 601 previously defined expression markers of the TAL1 stem-like cell population (Non-parametric Wilcoxon test, log2FC > 0.5, padj < 0.05) (Supplementary Methods, Fig. 2i and Supplementary Data 2). **e** PDX-derived cells of all patients ranked by stemness score. Natural breaks in the scores are identified by getJenksBreaks ($k = 4$). Cells with stemness score > 0.25 are considered stem-like cells in the follow-up analyses. **f, g** Stacked barplot displays fractions of stem-like cells vs other leukemic blasts: **f** per T-ALL subgroup. **g** per relapse type. **h** Stemness score in patient-derived cells of individual patients. Red dashed line reflects the threshold used for the definition of stem-like cells (0.25) that is based on Jenks natural break classification. Patients are grouped in three different categories based on the proportion of stem-like cells: Blue > 15%, green 5–15%, red < 5%. Created in BioRender. Costea, J. (2025) https://BioRender.com/5dwyagu. Source data are provided as a Source Data file.

(e.g., RPL32, RPL27A; detected in six and five patients), and epigenetic modifiers (e.g., BRD2; detected in three patients).

To understand the biological processes in which AS occurs in these cells, we performed Gene Ontology (GO) based over-representation analyses on these 161 genes, which revealed enrichment of AS in pathways presumed to contribute to the stemness phenotype (Fig. 4d) - in close agreement with and thus corroborating our aforementioned gene expression analyses. These include cell adhesion (padj < 0.01), stem-cell proliferation (padj < 0.05), and RNA metabolism (padj < 0.00001). In addition, as previously described in precursor B-ALL[40], we find that splicing factors themselves are also affected by AS (padj < 0.000001).

One of the recurrent events enriched in stem-like cells is a retained intron at node 10 and an additional exonic segment at node 11 of RPL27A which is shared among PDXs of four patients (P2, P6, P8, P12) (Fig. 4e, g and Supplementary Fig. 4e). The retention of these two nodes results in a premature stop codon and in an open reading frame for a non-coding protein (ENST00000530585) which is a likely target of nonsense mediated decay[43,44]. The finding of a likely inactivating alternative splicing event is compatible with the aforementioned downregulation of RNA translation and metabolism in the dormant population.

Next, we investigated the different splice isoforms of FOS, as FOS is one of the significant regulons in the stem-like cell population (Fig. 4f, h and Supplementary Fig. 4f). PDXs of patient P2, P8, and P10 show a higher inclusion of node 10 and 11 in their stem-like cells which, interestingly, leads to a shortened but functional protein isoform (ENST00000554617[43]). While the consequences of this shortened version on the activity of FOS is currently unknown, the critical function of FOS in stem cell biology[45] suggests a putative mechanism of mediating stem-like cell functions.

Together with the known impact of AS on the persistence of resistant subpopulations in hematologic malignancies[46], these data suggest that specific splicing isoforms may contribute to stemness and treatment resistance in T-ALL, although the direct validation of potential mechanisms governing such function remain beyond the scope of this study.

### Expansion of stem-like cell population during T-ALL evolution

Considering the quiescence phenotype and the downregulation of metabolic processes of the stem-like cells, we hypothesized that they may be particularly treatment resistant. We therefore analyzed this cell population from initial disease to relapse-based PDXs of those patients in whom the proportion of stem-like cells could be reliably quantified. Therefore, we selected 8 out of the 13 PDXs in whom stem-like cells contribute to at least 5% of the total cell population for further analysis (Fig. 5 and Supplementary Fig. 5). Stem-like cell clusters of each patient were identified based on the highest stemness score compared to other clusters of the same patient. At the time of initial diagnosis stem-like cells accounted for a very small proportion of all PDX cells (0.46%–2.16% in 6 patients) or remained undetectable in PDX-derived cells of 2 patients. In contrast, by the time of relapse, the cells with a stem-like cell phenotype accounted for a substantial proportion of the total cells ranging from 10.68% to 43.52% (Fig. 5d, paired $t$-test: $p = 6.6 \times 10^{-5}$). Remarkably, PDXs of four patients (P6, P1, P3, P7) display at least one more cluster exhibiting a high stemness score (cluster with more than 50% cells have a stemness score > 0.25), which is strongly enriched in the relapse sample. These data indicate that the stem-like cell population commonly expands during the transition from initial disease to relapse, potentially reflecting resistance to first-line therapy. By contrast, the initially predominant cells had largely disappeared in the relapse samples, which is consistent with the treatment response to first-line therapy largely resulting in low levels of residual disease or even transitory complete remission.

### Increased tolerance of T-ALL stem-like cell populations to conventional treatment and their potential implications for risk stratification

We next sought to validate the hypothesis that stem-like cells are less sensitive to treatment. We thus treated PDX-dervied cells obtained from three relapsing patients (P1, P6, and P10) in-vitro for three days with Cytarabine (a key component of current treatment protocols). These cells were subsequently analyzed by high-throughput single-cell transcriptomics (10× Genomics scRNA-seq), resulting in the additional analysis of 28,733 high-quality single cells (Supplementary Methods). The samples from all three patients showed a substantially higher stemness score following treatment (Fig. 6a: $p = 2.69 \times 10^{-61}$, Fig. 6b: $p = 2.27 \times 10^{-119}$, Fig. 6c: $p = 3.23 \times 10^{-15}$), indicating that treatment either caused the upregulation of stemness markers in T-ALL cells or the preferential survival of cells enriched in stemness.

To perform further validation, we subsequently analyzed T-ALL patient cells in a murine PDX in-vivo, using diagnostic PDX samples of two relapsing patients P1 and P10, which we re-engrafted and then treated in-vivo (Fig. 6d, e). We applied two different treatment protocols, using either Cytarabine or a combination of Vincristine, Doxorubicin and Dexamethasone (Supplementary Methods). After 7 days of treatment, samples were collected from the murine bone marrow and analyzed by scRNA-seq. Analysis of 30,174 high quality cells revealed that the stemness score is increased in both cases following in-vivo treatment with the single agent Cytarabine, similar to the results observed in-vitro, and also following the combination therapy more closely resembling the actual treatment protocols employed in pediatric T-ALL patients. We additionally investigated the potential of four different anti-apoptotic inhibitors – MCL1 inhibitor S-63845, BCL-6 inhibitor FX1, BCL2-inhibitor Venetoclax, and BCL-xL inhibitor A1331852–to overcome treatment resistance by comparing the overall cell viability between initial disease and relapse with >5% stem-like cells following in-vitro treatment in PDX-derived cells (Supplementary Fig. 6 and Supplementary Data 8). As expected, cell viability significantly increased in relapse compared to initial disease when cells were treated with Cytarabine (paired $t$-test, $p = 0.045$). This indicates

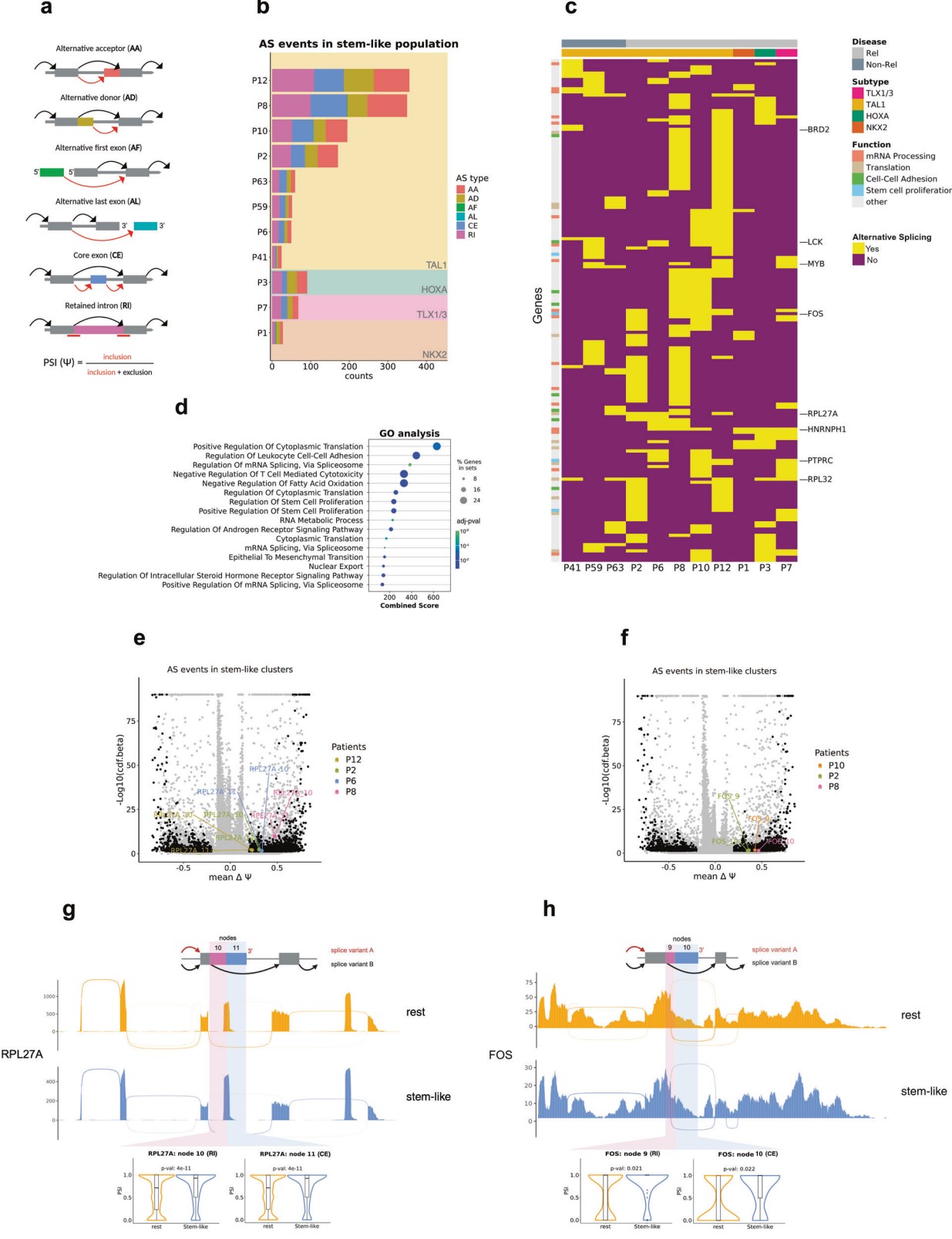

pronounced treatment resistance to chemotherapy in relapse, which can be at least partially explained by the expansion of treatment-resistant stem-like cells. In contrast, there was no significant difference in cell viability between initial disease and relapse observed when cells were treated with different anti-apoptotic inhibitors. However, further investigation is needed in in-vivo settings and in combination with

other therapeutic agents to fully evaluate their potential efficacy. Lastly, we applied our stemness score to a large RNA dataset of 1336 diagnostic T-ALL samples from a separate T-ALL patient cohort (Fig. 6f)[24] to investigate its potential as an additional biomarker for risk stratification. Intriguingly, our analysis reveals a higher stemness score in T-ALL of patients who were classified as M3 (> 25% of morphological

**Fig. 4 | The stem-like cell population presents unique AS variants. a** Illustration of all possible AS event types identified by the workflow (Supplementary Methods, Supplementary Fig. 4a). Psi (ψ) value refers to the number of reads supporting the inclusion of the splicing node. Created in BioRender. Costea, J. (2025) https://BioRender.com/abh7ogz. **b** Type and counts of detected AS events of PDXs from individual T-ALL patients with >5% stem-like cell population ($n = 19$ samples from 11 patients). Background colors categorize patients according to their T-ALL subgroup. **c** List of AS genes (rows) shared across stem-like populations of minimum two patients (columns). Yellow indicates that the gene presents AS in the stem-like cell cluster of the patient's PDX, purple indicates no significant difference. Patients are grouped based on their T-ALL subgroup and non-relapsing and relapsing disease. Left side of the panel highlights the biological pathways most affected by AS. **d** GO analysis of significantly enriched pathways (GO Biological Processes 2023) in PDX-derived samples ($n = 19$ samples from 11 patients) using all genes from **c**) ($n = 161$) as input. $P$-values were computed by one-sided hypergeometric test and adjusted for multiple testing (Benjamini-Hochberg). **e, f** volcano plot illustrates inclusion rates ($x$-axis) of splicing nodes in stem-like cell cluster. $Y$-axis represents significance values ($-$Log10(cdf.beta), see Supplementary Methods). All significant events are shown in black. **e** volcano plot shows splicing events of PDXs derived from four patients: P2, P6, P8, P12. Colors highlight two enriched splicing nodes (RPL27A node 10 and RPL27A node 11) in stem-like cell clusters of the four patients. **f** volcano plot shows splicing events of PDXs from three patients: P2, P8, P10. Colors highlight two enriched splicing nodes (FOS node 9 and FOS node 10) in stem-like cell clusters of PDX-derived samples from three patients. **g, h** Upper panel: schematic representation of two splicing variants observed in illustrated splicing nodes (see illustration in **a**). Red arrow indicates the splicing variant enriched in stem-like cells. Middle panel) Sashimi plots of stem-like cells (yellow) and other leukemic cells (blue) illustrate whole gene affected by alternative splicing. Read coverage is computed on P2, P6, P8, P12 (**g**) or P2, P8, P10 (**h**). Lower panel: Violin plot represents ψ value of affected splicing nodes in stem-like cell cluster (yellow) vs other leukemic cells (blue). $P$-values were computed using a two-sided Wilcoxon test. Boxplots depict the median as the central line, with the box edges marking the 25th and 75th percentiles. **g** Affected splicing nodes of RPL27A gene. Created in BioRender. Costea, J. (2025) https://BioRender.com/5thdqt7. **h** Affected splicing nodes of FOS gene. Created in BioRender. Costea, J. (2025) https://BioRender.com/1m4hb55. Source data are provided as a Source Data file.

blasts after 29 days of treatment) compared to patients classified as M2 (5–15% morphological blasts, $p = 0.011$) and patients classified as M1 ($< 5\%$ morphological blasts, $p = 8.6 \times 10^{-7}$)—thus demonstrating a significant association of a high stemness score with poor induction treatment response.

## Discussion

Understanding the mechanisms driving relapse and treatment resistance in T-ALL remains an unmet clinical need. While stem-like cells have been discussed as a potential type of cells with particular chemoresistance, the cellular networks underlying such resistance can only be incompletely explored by bulk sequencing. This limitation is aggravated, as stem-like T-ALL cells presently cannot be readily identified and isolated by standard techniques.

Harnessing VASA-seq[13], we provide comprehensive insights into the transcriptomic landscape of T-ALL from initial diagnosis to relapse, by unveiling a previously elusive subpopulation of T-ALL cells characterized by a quiescent, stem-like cell phenotype. We identified distinct transcriptional signatures, gene regulatory networks, and splicing isoforms, which we hypothesize to govern the biology and the treatment resistance of these stem-like cells.

Consistent with previous findings in precursor B-ALL, a large fraction of stem-like cells persisted in G1 phase, showing upregulation of stem-cell related pathways, such as cell adhesion known to facilitate the interaction with the stromal environment and resulting in the protection against external factors in the stem-cell niche[23,47]. The analysis of recurrent AS events revealed an enrichment in such cells in mRNA isoforms potentially impacting the metabolic activity, including those encoding ribosomal proteins (such as RPL27A), and those that are expected to modulate stem-cell induction, including FOS[45]. Indeed, AS might provide a post-transcriptional source for the regulation of stemness networks in T-ALL.

Mapping the expression profiles of the T-ALL PDX-derived samples to a thymic reference[16] assigned the majority of stem-like cells as DN progenitors. Our data thus indicate that T-ALL stem-like cells are particularly immature, which is consistent with the exquisite treatment resistance of this type of cells.

Our data imply that the clonal expansion of stem-like cells is more common in TAL1-driven T-ALLs, and particularly in type-2 T-ALL relapses, suggesting that early leukemic ancestors driving type-2 relapses might include stem-like cells as defined in our study. These data also align with the hypothesis that stem-like cells can derive from pre-leukemic clones in early phases of the disease[22,48,49]. We also identified stem-like cells in HOXA-, NKX2-, and TLX1/3-driven T-ALL PDX models, albeit at lower frequencies. Future research should further investigate the proportion of stem-like cells in non-TAL1 T-ALL subgroups to provide a more comprehensive understanding of their distribution. Additionally, exploring the potential impact of different treatment regimens on the emergence of stem-like cells during relapse will be essential to determine whether therapeutic strategies influence their frequency and characteristics.

The enhanced treatment resistance of this stem-like cell population was validated through both in-vitro and in-vivo drug-testing, highlighting the need for alternative strategies to target these resistant cells. One promising approach could be the use of anti-apoptotic inhibitors. This is supported by our in-vitro data, which indicate that treatment resistance is not increased in relapse compared when using anti-apoptotic inhibitors despite the expansion of the stem-like cell population in PDX samples of these patients. This finding highlights their potential in overcoming resistance in these persistent cell populations.

One potential limitation of our study is the use of PDXs rather than primary patient material, which could impact the direct translatability of some of our findings. However, prior analyses have demonstrated that these PDXs closely mirror the genetic and epigenetic landscapes of primary patient samples[50] reinforcing their reliability for studying T-ALL biology.

The presence of stem-like cells did not per se predict the occurrence of a relapse in our data. Yet, the stemness patterns identified in our study is predictive with treatment induction failure in a large publicly available RNA-seq dataset derived from primary T-ALL samples[24]. This suggests that conclusions drawn from our PDXs are relevant to patient outcomes, and can potentially serve as a useful biomarker for patient stratification.

Given the need for developing new treatment strategies, the stem-like cells defined in our study should be considered as future therapy targets to prevent relapse and overcome resistance. The stemness score developed in our study may therefore identify this resistant cell population, serving as a clinical biomarker.

## Methods
### Patients

The primary cells were obtained from patients recruited in ALL-BFM 2000, ALL-BFM 2009, CoALL03, and CoALL09 trials. Patients' clinical characteristics have been described previously[7]. All clinical trials from which samples were used in this analysis received prior approval from the relevant institutional review boards or ethics committees: the Ethics Committee of the Hannover Medical School for the ALL-BFM-2000 study (reference number AZ 2522), the

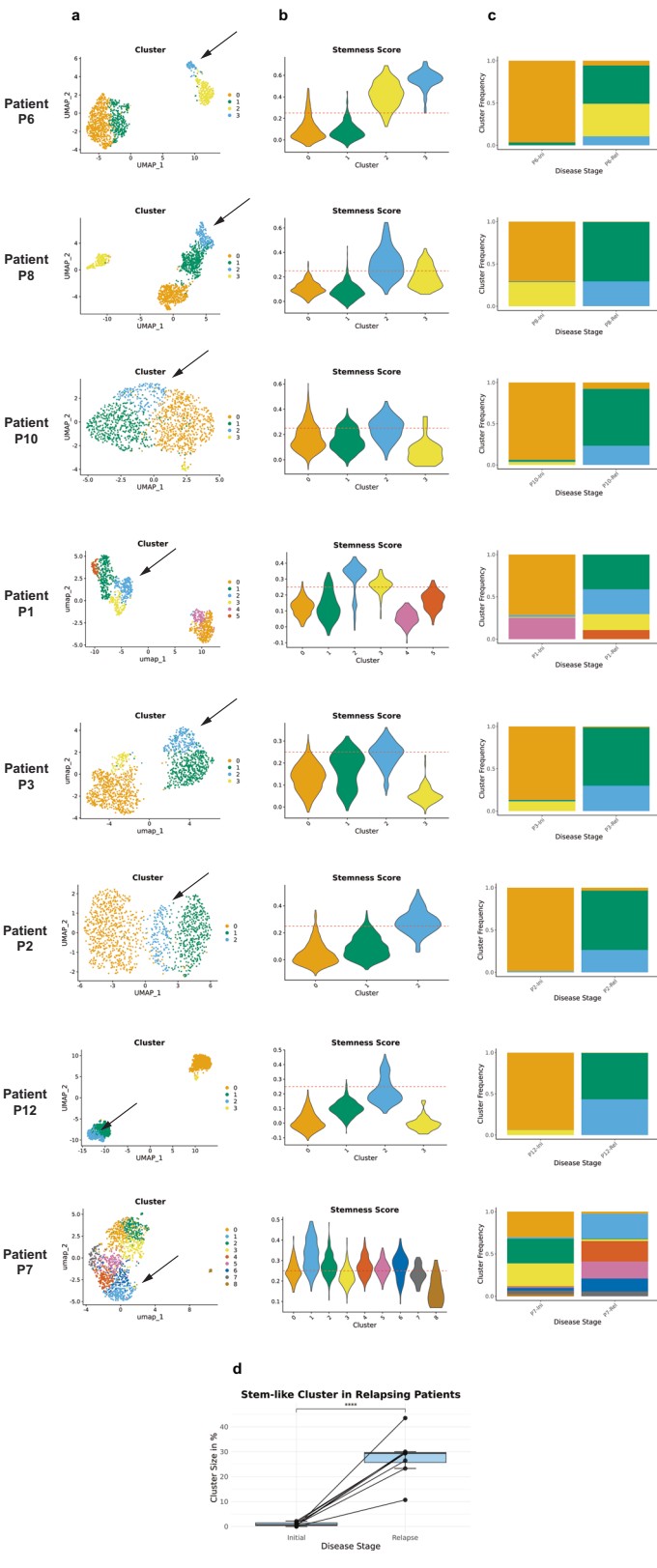

**Fig. 5 | Stem-like cells expand during the transition from initial diagnosis to relapses.** PDXs of 8 patients with >5% stem-like cells are shown, each derived from primary samples taken at initial diagnosis and relapse. **a** Clustering based on RNA expression performed for individual patients. Visualization as UMAP plots. Number of cells per UMAP: P6 (n = 1204 from two samples), P8 (n = 1276 from two samples), P10 (n = 1330 from two samples), P1 (1042 from two samples), P3 (n = 1355 from two samples), P2 (n = 1231 from two samples), P12 (n = 1298 from two samples), P7 (n = 1369 from two samples). Black arrow points towards the cluster with the highest stemness score. **b** Stemness score (see Fig. 3d) displayed per patients' individual clusters. Red dashed line reflects the threshold used for the definition of stem-like cells (0.25) (see Fig. 3e). **c** Stacked barplots display the frequency of clusters at the time of initial diagnosis and relapse in PDXs of individual patients. **d** Boxplot displays significant enrichment in the frequency of stem-like cells at the relapse compared to initial diagnosis in PDXs of all 8 patients with more than 5% stem-like cells in total. Boxplots show median (center line), interquartile range (box limits), whiskers extend to 1.5× IQR, and points represent outliers. Significance was evaluated using a two-sided paired t-test. Source data are provided as a Source Data file.

Supplementary Data 1. Samples from both male and female patients were included. Written informed consent had been obtained from all the patients or legal guardians, and the experiments conformed to the principles set out in the WMA Declaration of Helsinki and the Department of Health and Human Services Belmont Report.

### Patient-derived xenografts (PDXs)
We maintained T-ALL patient cells as PDXs as previously described[51]. In-vivo experiments were approved by the veterinary office of the Canton of Zurich, in compliance with ethical regulations for animal research.

### In-vitro drug treatment with Cytarabine
MSCs were seeded in 24-well plates at a concentration of 500.000 cells per well in 1 ml AIM V medium. After 24 h, cryopreserved T-ALL PDX cells were added at a concentration of 1.5 million cells per well in 1 ml AIM V. Cytarabine (MedChemExpress, HY-13605) or DMSO (vector) as control was added after an additional 24 h at a concentration of 1 μM. After 72 h cells were trypsinized, collected, and frozen in 90% FBS/ 10% DMSO.

### In-vivo drug treatment
Mice with a human engraftment above 5% measured through flow cytometry using mCD45 (eFluor 450, eBiosciences, 1:100), hCD7 (PE, eBiosciences, 1:25) and hCD45 (Alexa Fluor 647, BioLegend, 1:25) were started on treatment either with a combination of 0.50 mg kg⁻¹ of vincristine (1 mg ml⁻¹ injection, Teva Pharma) administered weekly intraperitoneally, 10.5 mg kg⁻¹ of dexamethasone (4 mg ml⁻¹, Mepha Pharma AG) administered daily intraperitoneally, and 2 mg kg⁻¹ of doxorubicin (2 mg ml⁻¹, Sandoz) administered weekly intravenously or a mono-treatment with 25 mg/kg Cytarabine (Cytosar solution 100 mg/5 ml, Pfizer). After 7 days of treatment the leukemia cells were collected by flushing the bone marrow. Untreated mice served as controls, and their leukemia cells were collected on the same day as those from the treatment groups.

### Single cell RNA sequencing
For VASA-seq, mCD45-DAPI- were sorted as single cells into cooled 384-well plates and libraries were prepared by the company according to the VASA-seq protocol[13] and sequenced on the Nova Seq X Plus (10B–100 cycle, paired-end, 150.000 reads/cell).

For 10× Genomics bulk of mCD45-DAPI- cells were sorted and samples were processed immediately according to standard 10× Genomics Chromium 3' (v.3.1 Chemistry) protocol. Libraries were

Ethics Committee of the Christian Albrechts University of Kiel for the AIEOP-BFM-2009 study (reference numbers A177/09) and the Ethics Committee of the Ärztekammer Hamburg for the CoALL-07-03 (reference numbers 2077) and for the CoALL-08-09 (reference number PVN3409) studies. To protect patient privacy, sex information has been omitted and age is presented in ranges in

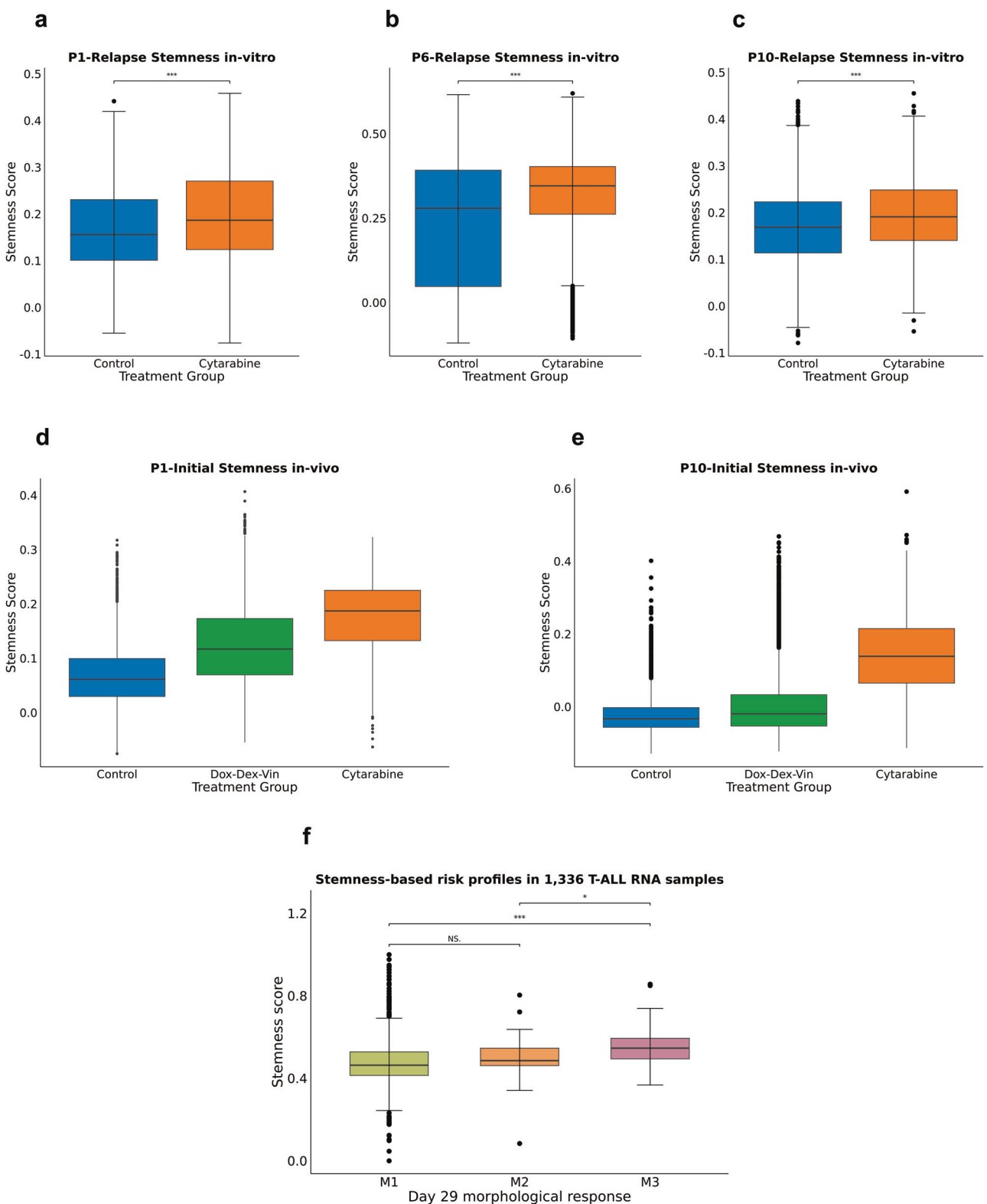

sequenced on the NovaSeq 6000 S2 v1.5 (100 cycles) flowcell S2 (Surface: 3.3-4.1 BIO reads).

### Permutation tests

We used the package "scProportionTest" ([https://github.com/rpoli castro/scProportionTest](https://github.com/rpolicastro/scProportionTest)) to analyze the difference in cell type and cell cycle proportions. Number of permutations per analysis were set to 1000. Significance threshold: Log2FoldDifference (Log2FD) > 0.25 and FDR < 0.05.

### Alternative splicing (AS) analysis

The pre-processing of the raw FASTQ files from VASA-Seq was performed via a custom Snakemake[52] pipeline. See Supplementary Methods for further details.

**Fig. 6 | Functional validation of treatment resistant stem-like cells. a–c:** in-vitro treatment of 28,733 PDX-derived cells from three relapsing patients (P1, P6, P10). T-ALL single cells have been analyzed by scRNA-seq. Boxplots display the stemness score of single cells treated with DMSO control (blue) and 1 μM Cytarabine (orange). Boxplots show median (center line), interquartile range (box limits), whiskers extend to 1.5× IQR, and points represent outliers. Significance was analyzed using a two-sided t-test. **d, e:** 30,174 PDX-derived cells of 2 relapsing patients (P1 and P10) at the time of initial diagnosis have been re-engrafted into mice followed by in-vivo treatment for 7 days. T-ALL single cells have been collected from the murine bone marrow and analyzed by scRNA-seq. Boxplots display the stemness score of single cells that have not been treated (blue) or otherwise treated with either a combination of doxorubicine (1 mg/kg weekly intravenously), vincristine (0.25 mg/kg weekly intraperitoneally) and dexamethasone (10.5 mg/kg daily intraperitoneally) (green) or else treated with Cytarabine (25 mg/kg daily intraperitoneally) as single agent (orange). **f** Stemness score analysis of a public bulk RNA dataset comprising 1336 T-ALL patients at initial diagnosis[24]. Patients have been grouped based on their morphological response after 29 days of treatment: M1 (light green) <5% blasts, M2 (light orange) 5–25% blasts, M3 (light purple) > 25% blasts. Boxplots show median (center line), interquartile range (box limits), whiskers extend to 1.5× IQR, and points represent outliers. Significance was analyzed using a two-sided t-test. Source data are provided as a Source Data file.

## Reporting summary

Further information on research design is available in the Nature Portfolio Reporting Summary linked to this article.

## Data availability

The publicly available RNA-seq data used in this study are available from the database of Genotypes and Phenotypes (dbGaP) under accession number phs002276.v2.p1[24]. The scRNA-seq data generated in this study have been deposited in the European Genome phenome Archive database under accession code EGAS50000000582. Due to the sensitive nature of human genomic data and to protect participant privacy, the data is access-controlled. Data access is governed by the Data Access Committee (DAC), which ensures compliance with ethical, legal, and institutional guidelines. Researchers seeking access must submit a data access request via the EGA platform, outlining the intended use of the data. Access will be granted to researchers affiliated with academic or research institutions and requests will typically be reviewed and responded to within 10 working days. Once approved, data will be available to the requestor until project completion. The remaining data are available within the Article, Supplementary Information or Source Data file.

## Code availability

The computational workflow related to the alternative splicing analysis can be found here: https://doi.org/10.5281/zenodo.15512143[53]. All other scripts related to the analysis of this study can be found here: https://doi.org/10.5281/zenodo.15514411[54].

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

## Acknowledgements

We thank all the patients who participated in the study and their families. We thank EMBL FCCF Services, EMBL IT Services, and EMBL GeneCore for technical support. We further thank D. Campana, St. Jude Children´s Research Hospital, Memphis, TN, for kindly providing MSCs. This work was supported by the Athenaeum Dietrich Götze Stiftung für Kultur und Wissenschaft, by the Mildred-Scheel-Postdoc Program (Deutsche Krebshilfe), as well as the Health + Life Science Alliance Heidelberg Mannheim, and received state funds approved by the State Parliament of Baden–Württemberg.

## Author contributions

J.C. designed and performed the scRNA-seq experiments, run bioinformatic analyses, and wrote the manuscript. K.K.R. established PDX models. K.K.R. performed in-vitro and in-vivo drug treatment assays. P.Z. performed splicing variant analysis and run pySCENIC pipeline. T.R. supervised computational analysis. A.M. helped with the set-up of the pySCENIC pipeline[14,15,55] and run initial pySCENIC analyses. J.Z. supervised the set-up of the pySCENIC pipeline[14,15,55]. M.S., G.E., and C.E. provided patients samples and data for the analyses. J.P.B. and B.B.R. supervised the establishment of the PDX models and drug treatment assays. A.E.K. and J.O.K. designed the research, supervised the project, and wrote the manuscript. All authors reviewed and contributed to the final manuscript.

## Funding

## Competing interests

The authors declare no competing interests.

## Additional information

[1]Molecular Medicine Partnership Unit (MMPU), EMBL and Medical Faculty of Heidelberg University, Heidelberg, Germany. [2]European Molecular Biology Laboratory (EMBL), Genome Biology Unit, Heidelberg, Germany. [3]Faculty of Biosciences, Heidelberg University, Heidelberg, Germany. [4]Division of Pediatric Oncology, University Children's Hospital, Zürich, Switzerland. [5]Molecular Biosciences/Cancer Biology Program, Heidelberg University and German Cancer Research Center (DKFZ), Heidelberg, Germany. [6]European Molecular Biology Laboratory (EMBL), Genomics Core Facility, Heidelberg, Germany. [7]European Molecular Biology Laboratory, Molecular Systems Biology Unit, Heidelberg, Germany. [8]Artificial Intelligence in Oncology, German Cancer Research Center (DKFZ), Heidelberg, Germany. [9]Department of Pediatrics, University Hospital Schleswig-Holstein, Campus Kiel, Kiel, Germany. [10]Department of Pediatric Oncology/Hematology, Charité Universitätsmedizin Berlin, Berlin, Germany. [11]German Cancer Consortium (DKTK), German Cancer Research Center (DKFZ), Heidelberg, Germany. [12]Clinic of Pediatric Hematology and Oncology, University Medical Center Hamburg-Eppendorf, Hamburg, Germany. [13]Department of Pediatric Oncology, Hematology, and Immunology, Heidelberg University, Heidelberg, Germany. [14]Hopp Children's Cancer Center (KiTZ) Heidelberg, Heidelberg, Germany. [15]Clinical Cooperation Unit Pediatric Leukemia, German Cancer Research Center (DKFZ), Heidelberg, Germany. [16]Bridging Research Division on Mechanisms of Genomic Variation and Data Science, German Cancer Research Center (DKFZ), Heidelberg, Germany. [17]These authors contributed equally: Julia Costea, Kerstin K. Rauwolf. ✉e-mail: andreas.kulozik@med.uni-heidelberg.de; jan.korbel@embl.de

