## [Transparent Peer Review file · Nature Communications]

Role of Stem-Like Cells in Chemotherapy Resistance and Relapse in Pediatric T-Cell Acute Lymphoblastic Leukemia

Corresponding Author: Ms Julia Costea

Version 0:

Reviewer comments:

Reviewer #1

(Remarks to the Author)

Costea et al investigate the stem cell like compartment of T-ALL using single cell total transcriptome (VASA-seq) of 13 matched diagnostic/relapse samples to establish a link between dormant stemness features and drug resistance. Overall, authors aim to provide a unifying theme to disease refractoriness, linking it to increased stem cell compartment at relapse. The concept is interesting, additional experimental evidence would further support the conclusions drawn, as detailed below. Very interesting, worth reporting.

General comments

Reference to the current genomic landscape of T-ALL should be added to introduction.

Figure parts could be presented by alphabetical order for clarity and to facilitate reading. Their numbering could be improved for clarity.

A comment to distinguish single cell full length RNAseq (VASA-seq) from single molecule long read sequencing should be added, to provide more detailed insight on this recently reported approach.

Please define "cancer predisposing genes" in introduction

Figure 1

An index case of relapsed T-ALL is presented with matched diagnostic/relapse VASA-seq characterization.

For this case as well as for the entire T-ALL cohort presented, additional information pertaining to patient characteristics (eg age, BM infiltration, CNS status, time to relapse, follow-up interval, etc), molecular subtypes including co-mutational profiling and immunophenotype should be provided, when available. A table with all patients analysed would provide clarity to the manuscript (in supplement section).

For cases of the TAL1 subtype, reference to the recently described T-ALL classification of alpha-beta like and DP-like subsets would enrich the manuscript (PMID 39143224).

Analyses are performed on in vivo expanded cells. How does the clonal distribution change between patient and PDX expanded cells, for the same sample? Is it similar or do you see a clonal selection in the PDX model? Why are single cells analyses performed on in vivo expanded cells vs primary patient samples? What mouse strain was used for PDX generation? What patient tissue was used for PDX generation (bone marrow?), and what mouse tissue was harvested for single analyses? These details could be provided in the supplemental data section.

For index case (P2), 1231 single cell full length transcriptomes were generated, including 653 for diagnostic sample and 578 for relapse sample. Can the authors comment on leukemia stem cell frequencies of T-ALL in relation to the numbers of single cell transcriptome performed?

Cell clusters were mapped to a thymic single cell reference to infer T-cell maturation stage. A similar strategy mapping clusters to normal hematopoietic stem/progenitor cell atlas would define stage of cell differentiation.

There is no reference to Fig. 1j in results section.

Figure 2

A set of 5 relapsed TAL1 driven T-ALL is compared to 5 non relapsing TAL1 T-ALL.

As suggested in Fig.1, please provide clinical and molecular characteristics of the samples.

Mapping to normal hematopoietic stem/progenitor atlas would contribute to define stage of differentiation.

Can you provide an estimate of the stem cell fractions in the five relapsing samples?

Fig. 2g: reference to TAL1 alpha-beta like and DP-like subsets could be added.

DGE analyses of stem clusters vs all cell clusters in relapsing T-ALL is presented.

Page 7: "which is characterized by low cell division and which we characterized in further detail in the following" please revise wording.

Figure 3

Stemness score across distinct molecular subtypes of T-ALL. Seven additional samples from 3 distinct T-ALL genetic subtypes are included.

Why was TAL1 case P11 excluded from Fig.2?

Given the molecular heterogeneity of T-ALL, this reviewer is concerned that the study is not sufficiently powered to conclude that stem-like cells are more prevalent in TAL1 driven ALL, vs other T-ALL genotypes. A focus on the TAL1 T-ALL may be sufficient.

Are the authors trying to link type II relapse to a higher fraction of stem-like cells? This is an interesting concept that potentially can be brought forward with the number of samples analyzed.

Fig. 3g not easy to understand as it stands.

A comment in the discussion should try and reconcile the fact that the presence of a stem-like population does not per se predict the risk of relapse.

Figure 4

Patterns of unique alternative splicing variants in stem-like cells.

Are AS variant also detected in the non-stem cell clusters? Are AS patterns distinct in stem vs non-stem clusters.

Are AS variants involving RPL27A restricted to the TAL1 subgroup? In itself, this would be an interesting finding.

In figure legend 4c, do authors mean purple or black?

Figure 5

Expansion of stem-like cell clusters with progression to relapse.

Are all 8 patients datasets performed at relapse? Please specify in figure 5 legend.

What distinguishes the stem cell clusters within an individual patient?

P6, P1, P3, P7 are of distinct molecular subtypes, P6 is a TAL1 T-ALL. Is a stem cell expansion noted in all TAL1 relapse samples?

Figure 6

Overall, the functional data is limited and presented in a rather unconventional way.

Why was P41 selected for in vivo functional testing? This is a non-relapsing TAL1 patient.

Distinct cell types and are used for in vitro and in vivo functional testing, which limits the conclusions that can be drawn, including impact of in vivo resistance from the niche.

In vitro cell culture conditions should be detailed.

In terms of drug treatment selection, authors note upregulation of pro-survival factors BCL-2 and MCL1 in the stem-like clusters, as well as anti-apoptotic signalling pathways. Showing the response of stem-like clusters to BCL-2 or MCL1 inhibitors would provide significant pre-clinical data, given the use of pro-survival factors inhibitors in the context hematological malignancies. Whether stem like cells are resistant or sensitive, this would provide meaningful functional data, in this reviewer opinion.

Discussion should address some of the concerns raised, such as correlations of stem-like clusters to T-ALL molecular subtypes, use of PDX models vs primary patient cells, T-ALL stem cell frequencies in relation to the numbers of cells sequenced using VASA-seq. Perhaps a more restricted focus on the TAL1 subset is to be considered.

Reviewer #2

(Remarks to the Author)

The authors aim at identifying and characterizing a stem-like population in T-ALL that could be responsible of resistance to chemotherapy. The paper is very well written and very easy to follow even for non T-ALL experts. It is also very commendable that the authors used VASA-Seq rather than the (overused) 3' 10X as it has provided them with the capacity to properly delve into stemness-specific alternative splicing patterns. However, in my opinion, the computational analyses have several major flaws.

1) Some wording regarding the analysis have to be changed as they are incorrect. UMAP is not an analysis method, it is just a representation in 2D of high-dimensional data. Moreover, there are recent publication showing that the "structure" of the UMAP plot is meaningless, and as such, it is not an "actionable" space and should be use with lots of caution (e.g., PMID: 37590228). Therefore, sentences such as "Uniform Manifold Approximation and Projection (UMAP) analysis" should be rephrase.

2) I do not understand well the rationale to perform the gene expression analysis first on one unique patient, and then see how it replicate in other patients through SCENIC (what about if integrating TAL1 patients via their expression?), to then see how it replicate in other type of ALLs. My main concern here is that the authors may be overfitting to a stem-like phenotype appearing in that particular patient (or the TAL1 subtype) and uncovering signatures that are more related to TAL1 T-ALL than to T-ALL.

3) It is not very clear how the different analyses were performed. While the Methods section and the Supp. Method section shed some light into them, there is no link to the scripts, and hence I cannot asses how the analyses were actually performed. On these lines, were the samples from pre and post relapse in P2 integrated in the first section? If not, how do you know that the seen differences are not due to a batch effect?

4) Also, it is not clear how genes were ranked/selected for the GSEA. Were ranked by expression? or ranked by LFC between stem-like and blasts? This is important to clarify further.

5) When computing the stemness score, the authors should compare their analysis to using gene expression directly. Again, I am not sure why authors did not perform traditional gene expression analysis. Also, the authors should consider using SCENIC+ (or even DeepSCENIC) which are updated versions of SCENIC (which is from 2017).

6) Regarding the alternative splicing section, I think this section adds a big value to the study and it is differential to what has been done before. However, the flaws in the computational identification of the stem-like cells downgrades a bit relevance of the section.

In summary, I believe that the analyses performed to identify the stem-like population are confusing, the computational analysis is not well presented, and the mix of gene expression analyses with SCENIC analyses across the different cohorts looks like cherry-picking. These precludes me to fully believe that the identified population is indeed a stem-like population associated to T-ALL, rather than a specific population that *looks like* stem-cells that are found in specific patients.

Reviewer #3

(Remarks to the Author)

The manuscript by Costea et al reports that majority of pediatric T-ALL patients exhibit stem-like subpopulation, which although is a small percentage at initial diagnosis, expands significantly after relapses and is resistance to therapy. The authors performed substantial scRNA-sequencing (VASA-seq) to carefully characterize the gene expression, splicing pattern, and TFs. The main take home message is that resistant stem-like cells are resistant to treatment thus persist and expand; and these cells can be found in all molecular subtypes of T-ALL tested (although mostly in TAL1+ samples). Since the leukemia stem cell population is not well-defined in T-ALL, this study provides important insights into the specific regulon program and the molecular properties of T-ALL stem-like cells. Overall, the experimental data are solid and compelling. However, important functional experiments are missing to support the main conclusion.

The major and minor points are listed below.

Major Points

1). What are the treatment histories of the relapsed patients used for scRNA-seq? Depending on the treatment, different "stem-like" cells may emerge based on genetic or epigenetic rewiring. It will be interesting to compare the stemness score and regulon patterns based on different treatment.

2). The experimental approach in Fig. 6a-b is problematic. First, the patient selection for in vitro cytarabine treatment is peculiar. Why treat relapsed patients to test if stem-like cells expand since they already expanded? Samples at initial

diagnosis prior to relapse will serve much better choice. Second, can authors explain why they choose patient p1 and p6? They don't have the highest stemness score (Fig. 3f). Lastly, for the in vivo treatment of p41 (Fig. 6e-h), it is important to show the expansion of stem-like cells are not simply the result of selective advantages of engrafting capacity. What's the cell composition in PDX without any treatment?

3). Can the UMAPs and boxplots comparing diagnosis and relapse shown for all relapsed patients rather than only the 8 samples with >5% stem-like cells? Since type I relapse (p11, p5, p9, and p4) generally has less stem-like cells, it will be interesting to see which subpopulation expands after treatment in these patients.

Minor Points

1). The panels g and f in Fig. 3 are flipped.

2). We suggested to move Supplemental Fig. 2d to main Fig 3, which will help to interpret the data with pt #.

3). How is the "stemness score" defined? Is it based on the regulon (TF) expression? This concept is frequently used to identify the stem-like cells (or stem-like) throughout the text. However, the exact method used to calculate the score cannot be found in supplemental method as mentioned by the authors.

4). Please provide a patient characteristic table.

Version 1:

Reviewer comments:

Reviewer #2

(Remarks to the Author)

I thank the authors for the effort in responding all my concerns. I especially appreciate the independent validation on a large cohort, which make the results significantly more robust. I also highly appreciate the availability of the code, and how well organized and commented is.

Regarding my comments on using P2 first and then seeing finding replication on the rest of TAL1+ T-ALL patients, and, in lesser extend, in T-ALL patients has been addressed. Although, as a last request, I would like to see the patient distribution across clusters in the gene-expression-integrated clustering shown in Figure 1 of the response to author letter. In this regard, I would include that Figure as supp. figure to show that "orthogonal" analysis on the data is also able to identify the stemness population.

Finally, I understand the argument about the potential "oversmoothing" of gene-expression integration methods, and the capabilities of GRN-driven models to "overcome" batch effects. While not requested for this work, I wanted to point out to the authors recent population-level integration models that may be of use for the future: <https://www.nature.com/articles/s41592-023-02035-2>; <https://www.biorxiv.org/content/10.1101/2022.10.04.510898v2>; <https://www.nature.com/articles/s41587-023-01940-3>

(Remarks on code availability)

The code is very well organized and well commented.

Reviewer #3

(Remarks to the Author)

The manuscript has improved significantly. In particular, the new in vivo data (Fig. 6) has proved that stem-like population arises during drug treatment and is more resistance. Moreover, the application of the "stemness score" in a large cohort of T-ALL patients further validated these findings. Lastly, the method and supplemental section is more complete and much easier to follow in the revised version.

(Remarks on code availability)

Reviewer #4

(Remarks to the Author)

Costea et al. investigate the mechanisms underlying T-ALL relapse through single-cell RNA sequencing on 13 matched pediatric T-ALL PDX samples at diagnosis and relapse, along with five non-relapsing PDX samples collected at diagnosis. Interestingly, 11 of 18 "patients" exhibited a subpopulation of T-ALL cells with a "stem-like" expression signature and splicing pattern. This subpopulation expanded substantially at relapse, indicating resistance to therapy. Increased "stemness" at diagnosis was associated with a higher risk of relapse. Chemotherapy resistance was validated through in vitro and in vivo drug testing.

This is a very interesting study. It is well written and significantly improved in its extensively revised version. The authors also provided a well-thought-out rebuttal letter to address the Reviewers' comments on the previous submission.

Since the T-ALL stem cell population is not well-defined, this study provides potentially useful information by defining a specific regulon program/transcriptomic signature of T-ALL "stem-like" cells. Overall, the experimental data are solid and convincing. The revised version somewhat improved the functional validation, which was the main weak point of the original submission.

One problem with the present submission is that, throughout the manuscript, PDX are referred to as "patients". This is misleading and should be corrected. Although the Authors state that in a previous study (surprisingly) they did not observe clonal selection of genetic/ epigenetic differences between patients and PDX, primary samples ex vivo from patients are not the same as PDX.

(Remarks on code availability)

Reviewer #5

(Remarks to the Author)

(Remarks on code availability)

POINT-BY-POINT REPLIES TO THE REVIEWER COMMENTS

Reviewer #1, expertise in pediatric ALL models (Remarks to the Author):

Costea et al investigate the stem cell like compartment of T-ALL using single cell total transcriptome (VASA-seq) of 13 matched diagnostic/relapse samples to establish a link between dormant stemness features and drug resistance. Overall, authors aim to provide a unifying theme to disease refractoriness, linking it to increased stem cell compartment at relapse. The concept is interesting, additional experimental evidence would further support the conclusions drawn, as detailed below. Very interesting, worth reporting.

Our response: We are thankful for your enthusiastic recognition of our work. We value the constructive and insightful feedback provided, based on which we have revised our manuscript.

General comments

1. Reference to the current genomic landscape of T-ALL should be added to introduction.

Our response: Thank you for the suggestion. We have provided more details on the current knowledge on the genomic landscape of T-ALL in the revised introduction.

2. Figure parts could be presented by alphabetical order for clarity and to facilitate reading. Their numbering could be improved for clarity.

Our response: We agree, and have improved the manuscript accordingly.

3. A comment to distinguish single cell full length RNAseq (VASA-seq) from single molecule long read sequencing should be added, to provide more detailed insight on this recently reported approach.

Our response: Thank you for the suggestion. We have now provided such a comment. Since space in the main text will be restricted, we have opted to detail the distinction in the supplementary methods.

4. Please define “cancer predisposing genes” in introduction

Our response: We thank the reviewer for this comment, and have included a sentence in the introduction focusing on the role of somatic mutations in germline cancer predisposing genes in the context of type-2 relapses.

Figure 1: An index case of relapsed T-ALL is presented with matched diagnostic/relapse VASA-seq characterization.

5. For this case as well as for the entire T-ALL cohort presented, additional information pertaining to patient characteristics (eg age, BM infiltration, CNS status, time to relapse, follow-up interval, etc), molecular subtypes including co-mutational profiling

and immunophenotype should be provided, when available. A table with all patients analysed would provide clarity to the manuscript (in supplement section).

Our response: We have now provided a supplementary table (Supplementary Table 1) with this information.

6. For cases of the TAL1 subtype, reference to the recently described T-ALL classification of alpha-beta like and DP-like subsets would enrich the manuscript (PMID 39143224).

Our response: We thank the reviewer for this comment, and we agree. We have now compared our TAL1 patients to the TAL1 alpha-beta like and DP-like subsets from Pölonen et al., Nature 2024 (PMID 39143224) (Suppl. Fig 2d-e).

7. Analyses are performed on in vivo expanded cells. How does the clonal distribution change between patient and PDX expanded cells, for the same sample? Is it similar or do you see a clonal selection in the PDX model? Why are single cells analyses performed on in vivo expanded cells vs primary patient samples? What mouse strain was used for PDX generation? What patient tissue was used for PDX generation (bone marrow?), and what mouse tissue was harvested for single analyses? These details could be provided in the supplemental data section.

Our response: We thank the reviewer for these insightful comments. Our previous studies demonstrated that our PDX samples closely mimic the genetic and epigenetic composition of primary T-ALL (Richter-Pechańska et al, EMBO 2018, PMID: 30389682) – without substantial changes in clonal composition. We now refer to this prior study in our revised manuscript. For clarification, a PDX model has been used for practical reasons, since primary samples for this rare disease entity are not available anymore unfortunately – these have been used for the analysis comparing PDX vs. primary samples mentioned earlier. Primary material has been taken from the patients' bone marrow and engrafted material has been taken from the murine spleen to perform the single cell analyses. We have added this information to the supplementary table referred to in comment #5. Cells have been engrafted in NSG (NOD.Cg-PrkdcscidII2rgtm1Wjl/SzJ) mice (as stated in the Supplementary Methods).

8. For index case (P2), 1231 single cell full length transcriptomes were generated, including 653 for diagnostic sample and 578 for relapse sample. Can the authors comment on leukemia stem cell frequencies of T-ALL in relation to the numbers of single cell transcriptome performed?

Our response: For clarification, we identified $N=9$ stem-like cells at initial disease and $N=153$ stem-like cells at relapse. We have added this information to the revised manuscript in response to this reviewer comment.

9. Cell clusters were mapped to a thymic single cell reference to infer T-cell maturation stage. A similar strategy mapping clusters to normal hematopoietic stem/progenitor cell atlas would define stage of cell differentiation.

Our response: Thank you for this insightful suggestion. We chose to map the cell clusters to a thymic single-cell reference, because T-ALL originates from progenitor T cells in the thymus, making it therefore the most relevant reference for our study. However, we agree that mapping to a hematopoietic stem/progenitor cell atlas could

provide valuable insights, particularly for understanding stem-like behavior within these cells. We therefore mapped our T-ALL dataset against a bone marrow reference dataset (Supplementary Fig. 1c-d, Supplementary Fig 2g-h) (Hao et al, 2021 Cell, PMID: 34062119). This analysis did not indicate that these cells derive from an earlier hematopoietic stage beyond the lymphoid lineage. Instead, stem-like cells resembled CD4 memory cells, a population characterized by pronounced dormancy, while the majority of cells in the other leukemic subpopulations resembled common lymphoid progenitors.

10. There is no reference to Fig. 1j in results section.

Our response: We apologise for the omission and now refer to this figure in the revised main text.

Figure 2: A set of 5 relapsed TAL1 driven T-ALL is compared to 5 non relapsing TAL1 T-ALL.

11. As suggested in Fig.1, please provide clinical and molecular characteristics of the samples.

Our response: We agree, and now refer to these data in Supplementary Table 1 (see our reply to comment #5).

12. Mapping to normal hematopoietic stem/progenitor atlas would contribute to define stage of differentiation.

Our response: We agree, and refer to comment #9 for our reply.

13. Can you provide an estimate of the stem cell fractions in the five relapsing samples?

Our response: Details on the stem cell fractions can be found in Figure 5, including for the 5 relapsing TAL1 patients, which based on our understanding the reviewer is referring to in this comment.

14. Fig. 2g: reference to TAL1 alpha-beta like and DP-like subsets could be added.

Our response: Thank you for your suggestion. As alluded to above, we have compared our cell populations to the TAL1 alpha-beta like and DP-like subsets from Pölönen *et al.*, *Nature* 2024 (PMID 39143224) (Suppl. Fig 2d-e).

15. Page 7: “which is characterized by low cell division and which we characterized in further detail in the following” please revise wording.

Our response: We rephrased this sentence, as suggested by the reviewer.

Figure 3: Stemness score across distinct molecular subtypes of T-ALL. Seven additional samples from 3 distinct T-ALL genetic subtypes are included.

16. Why was TAL1 case P11 excluded from Fig.2?

Our response: To clarify, in our initial analysis, we focused specifically on type-2 relapsing patients, which is why P11 was excluded from Fig. 2, with P11 being the

only TAL1+ sample that exhibits a type-1 relapse. We have clarified the reasoning behind the exclusion from Fig. 2 in the revised manuscript.

17. Given the molecular heterogeneity of T-ALL, this reviewer is concerned that the study is not sufficiently powered to conclude that stem-like cells are more prevalent in TAL1 driven ALL, vs other T-ALL genotypes. A focus on the TAL1 T-ALL may be sufficient.

Our response: We provide a comprehensive reply to this question below, in response to comment #18.

18. Are the authors trying to link type II relapse to a higher fraction of stem-like cells? This is an interesting concept that potentially can be brought forward with the number of samples analyzed.

Our response: Thank you for your comment. Our analysis reveals that the fraction of stem-like cells is significantly higher in TAL1+ patients when compared to all other subgroups (as shown in Fig. 3f and Supplementary Fig. 3a), and in type-2 relapses compared to type-1 relapses (see revised Fig. 3g and Suppl. Fig3b). It appears that the association between stem-like cell frequency and relapse type is even more pronounced than the association of stem-like cells and molecular subgroups – nonetheless both comparisons show significant differences. Nonetheless, we agree that in the future a more comprehensive analysis in non-TAL1 T-ALL patients will be needed, to clarify the proportion of stem-like cells in each subgroup. We now emphasise this aspect in our discussion. After careful consideration, we decided to keep the current subgroup comparison in the main text. This allows us to offer a thorough perspective, yet leaves space for future research to establish precise frequency ranges of stem-like cell proportions in each subgroup.

19. Fig. 3g not easy to understand as it stands.

Our response:

In this plot we have analyzed the activities of the significantly up and downregulated stem-like regulons of the TAL1 patients (Fig 2j) in individual patients of all subgroups. The activity is visualized by the *DotPlot()* function from the Seurat package. The size of the dot encodes the percentage of cells per patient, while the color encodes the average regulon activity level across all cells of the patient (yellow is high, purple is low). Patients with the highest fraction of stem-like cells (>15%) showed a higher activity of upregulated stem-like regulons, than patients with a moderate (5 - 15%) or low (<5%) stem-like cell proportion. In addition, patients with the highest fraction of stem-like cells, also show a lower activity of downregulated stem-like regulons than patients with a moderate or low stem-like cell proportion. We have reworked the Figure caption to ensure that this display item can be more readily understood by readers. But we agree that this figure is difficult to understand and in addition, might not add sufficiently important information to be included in the main section of the manuscript. However, this figure reconfirms the activity of regulons seen in TAL1 patients, in other patients and we have therefore decided to move it to the Supplements (Suppl. Fig 3d).

20. A comment in the discussion should try and reconcile the fact that the presence of a stem-like population does not per se predict the risk of relapse.

Our response: We have incorporated the observation that the presence of stem-like cells does not inherently predict the risk of relapse as a discussion point in our revised Discussion section, as recommended. However, we would like to highlight an additional analysis we performed during paper revision, summarized below in response to comment #36. Specifically, our findings show that the stemness score is significantly enriched in the bulk-transcriptome of patients with treatment induction failure, when analysing an independent large cohort of T-ALL patients (i.e. the cohort data recently published by Pölönen et al. Nature 2024; PMID: 39143224). This suggests that analysis of the stem-like population uncovered in our study could serve as a clinically valuable predictive tool. We now emphasise this important finding in our revised manuscript, including in the revised abstract.

Figure 4: Patterns of unique alternative splicing variants in stem-like cells.

21. Are AS variant also detected in the non-stem cell clusters? Are AS patterns distinct in stem vs non-stem clusters.

Our response: To clarify, we compared the frequency of different splicing variants between the following two groups: stem-like and non-stem-like T-ALL cells. As we do not have a "reference" splicing pattern, our analysis focuses on the relative enrichment of splicing events between these two groups. The splicing patterns that we refer to are significantly more common in the stem-like cell population when compared to those in non-stem-like cells, indicating a distinct splicing profile associated with stem-like clusters. We have clarified this in the revised manuscript.

22. Are AS variants involving RPL27A restricted to the TAL1 subgroup? In itself, this would be an interesting finding.

Our response: Thank you for your question and observation. Yes, we did observe alternative splicing (AS) events affecting RPL27A (specifically splicing node 10 and node 11) only in TAL1 patients. However, likely due to the limited number of patients at our disposal for the AS analysis (n=11), these findings do not currently reach statistical significance when applying Fisher's exact tests.

23. In figure legend 4c, do authors mean purple or black?

Our response: We intended to use purple in Figure 4c; however, we acknowledge that the previously chosen colour may not be the most effective. We have therefore made the necessary adjustments and changed the colour tone to a clear purple.

Figure 5: Expansion of stem-like cell clusters with progression to relapse.

24. Are all 8 patients datasets performed at relapse? Please specify in figure 5 legend.

Our response: Figure 5 presents samples collected at the time of initial diagnosis and at relapse for all eight patients shown. We have clarified this in the revised Figure legend.

25. What distinguishes the stem cell clusters within an individual patient?

Our response: Thank you for your question. To clarify, our study focuses on identifying shared biological characteristics of stem-like cells across T-ALL patients, as this approach lays the groundwork for establishing a fundamental definition of stem-like cells in T-ALL and for informing potential future clinical applications targeting these stemness networks. However, in order to also provide an individual-level perspective, Figure 1 presents a detailed analysis of patient P2.

26. P6, P1, P3, P7 are of distinct molecular subtypes, P6 is a TAL1 T-ALL. Is a stem cell expansion noted in all TAL1 relapse samples?

Our response: Among the TAL1 T-ALL patients, we observed that 5 out of 6 patients exhibited a stem-like cell frequency above 5%. All of them experienced a significant enrichment in relapsed samples. These 5 patients experienced type-2 relapses. In contrast, the only TAL1 patient who did not reach a stem-like frequency above 5% was P11, who later developed a type-1 relapse. We have clarified this in the main text.

Figure 6:

27. Overall, the functional data is limited and presented in a rather unconventional way. Why was P41 selected for in vivo functional testing? This is a non-relapsing TAL1 patient.

Our response: We thank the reviewer for this feedback. We agree that relapsing patients could provide more valuable information, and thus we have now performed *in-vivo* functional testing on two relapsing patients. This question is partially overlapping with comment #40 from reviewer #3, and we therefore provided a combined response below.

28. Distinct cell types and are used for in vitro and in vivo functional testing, which limits the conclusions that can be drawn, including impact of in vivo resistance from the niche.

Our response: We believe the reviewer may be referring to differences in cell clusters/populations rather than cell types. To improve generalizability and facilitate cross-patient comparisons and comparison among *in-vitro* and *in-vivo* conditions, we have revised our approach by focusing on stemness score differences rather than differences of individual cell clusters. This adjustment allows for a more standardized assessment across different patient samples and conditions.

Additionally, we recognize that there may be an omission in the reviewer's statement after "cell types and " possibly implying that different patient samples were used for *in-vitro* and *in-vivo* experiments. To better align our *in-vitro* and *in-vivo* data, we have indeed revised our patient selection strategy. For *in-vivo* experiments, we now focus on initial diagnosis samples of P1 and P10, as these had the highest fraction of stem-like cells at diagnosis and were part of the relapsing cohort. For *in-vitro* experiments, we expanded our analysis to include the P10 relapse sample in addition to P1 relapse and P6 relapse, ensuring that the same patients are represented in both *in-vitro* and *in-vivo* settings. While these samples represent different disease stages (initial diagnosis for *in-vivo* vs. relapse for *in-vitro*), we prioritized the *in-vivo* findings

as they provide a more physiologically relevant context. Our previous *in-vitro* data already demonstrated consistent enrichment of stem-like cells following treatment, and given the clear *in-vivo* results, we decided not to further expand the *in-vitro* experiments to also include initial samples.

For details on these changes, please refer to our response in comment #40.

29. In vitro cell culture conditions should be detailed.

Our response: For the scRNA-seq analysis of *in-vitro* drug treated T-ALL cells: MSCs were seeded in 24-well plates at a concentration of 500,000 cells per well in 1 ml AIM V medium. After 24 hours, cryopreserved T-ALL PDX cells were thawed and added at a concentration of 1.5 million cells per well in 1 ml AIM V. Cytarabine (MedChemExpress, HY-13605) or DMSO (vector) as control was added after an additional 24 hours at a concentration of 1 μ M. After 72 hours cells were trypsinized, collected and frozen in 90% FBS/10% DMSO. This information can be found in the methods section of our revised manuscript.

For the drug response profiles: Drug responses were assessed in T-ALL cell co-cultures on hTERT-immortalized primary bone marrow MSCs in 384-well plates (Greiner, REF781090). 2,500 MSCs per well were plated in 20 ml AIM V medium 24 hours before adding 10,000 T-ALL cells per well in 20 ml of AIM V recovered from cryopreserved PDX samples. After 24 hours different drugs were added by using Echo 650 Series Liquid Handler (Beckman Coulter). The tested compounds included Cytarabine, Venetoclax, S-63845, A1331852 and FX1 (all from MedChemExpress), which were applied in a concentration range from 0.1 nM to 10,000 nM. After incubating the cells for 72 hours, they were stained with CyQuant and imaged using the Operetta CLS (PerkinElmer), a high-content imaging system. The captured images were processed using BIAS (Single-Cell-Technologies), an analytical software tool. Drug response parameters were determined using Non-Linear Least-Squares Minimization and Curve-Fitting for Python, a statistical method used to model dose-response relationships.

This information can be found in the supplementary methods section of our revised manuscript.

30. In terms of drug treatment selection, authors note upregulation of pro-survival factors BCL-2 and MCL1 in the stem-like clusters, as well as anti-apoptotic signalling pathways. Showing the response of stem-like clusters to BCL-2 or MCL1 inhibitors would provide significant pre-clinical data, given the use of pro-survival factors inhibitors in the context hematological malignancies. Whether stem like cells are resistant or sensitive, this would provide meaningful functional data, in this reviewer opinion.

Our response: We thank the reviewer for this great comment. In response, we performed *in-vitro* drug response profiling using anti-apoptotic inhibitors (Venetoclax for BCL-2 inhibition, FX1 for BCL-6 inhibition, S-63845 for MCL1 inhibition and A1331852 for BCL-xL inhibition) on samples from relapsing patients at both initial diagnosis and relapse. We measured cell viability between these disease stages after 72h of treatment. Drug response parameters like logAUC were calculated using a Non-Linear Least-Squares Minimization and Curve-Fitting for Python. Our new data

revealed that resistance to venetoclax (BCL-2 inhibitor), FX1 (BCL-6 inhibitor), S-63845 (MCL1 inhibitor) and A1331852 (BCL-xL inhibitor) does not increase at relapse. In contrast, with cytarabine treatment we detected a significant increase in resistance in relapse samples compared to diagnosis samples (see Supplemental Fig. 6).

31. Discussion should address some of the concerns raised, such as correlations of stem-like clusters to T-ALL molecular subtypes, use of PDX models vs primary patient cells, T-ALL stem cell frequencies in relation to the numbers of cells sequenced using VASA-seq. Perhaps a more restricted focus on the TAL1 subset is to be considered.

Our response: We thank this reviewer for this constructive comment. Accordingly, we have expanded our discussion to address the correlation of stem-like clusters with T-ALL molecular subtypes, the use of PDX models versus primary patient cells, and T-ALL stem cell frequencies in relation to the number of cells sequenced with VASA-seq. While we acknowledge that stem-like cells are particularly prevalent within the TAL1 subset, our data suggest that they are not confined to this subgroup, and thus in our view a broader perspective including the initial observations our study made in other subgroups is worth including in our manuscript. As stated above, our revised Discussion now highlights that more comprehensive analyses in other subgroups need to be undertaken in the future to clarify stem-like cell frequency ranges per subgroup.

**Reviewer #2, expertise in scRNA-seq for hematological cancers and stem cells
(Remarks to the Author):**

The authors aim at identifying and characterizing a stem-like population in T-ALL that could be responsible of resistance to chemotherapy. The paper is very well written and very easy to follow even for non T-ALL experts. It is also very commendable that the authors used VASA-Seq rather than the (overused) 3' 10X as it has provided them with the capacity to properly delve into stemness-specific alternative splicing patterns. However, in my opinion, the computational analyses have several major flaws.

Our response: We thank the reviewer for the thoughtful review and overall positive comments regarding manuscript clarity and our choice of sequencing methodology. At the same time, we acknowledge the concerns regarding the computational analyses and have carefully considered the feedback provided to address any potential shortcomings of the methods applied. In the following comments, we have also provided explanations for our methodological choices, and have clarified any changes made to our manuscript to address specific reviewer questions.

32. Some wording regarding the analysis have to be changed as they are incorrect. UMAP is not an analysis method, it is just a representation in 2D of high-dimensional data. Moreover, there are recent publication showing that the "structure" of the UMAP plot is meaningless, and as such, it is not an "actionable" space and should be use with lots of caution (e.g., PMID: 37590228). Therefore, sentences such as

"Uniform Manifold Approximation and Projection (UMAP) analysis" should be rephrase.

Our response: We appreciate this comment and the guidance on the correct terminology regarding UMAP. In response, we have rephrased the respective sentences to clarify that our clustering approach is a graph-based clustering analysis performed using the Louvain algorithm (indeed, UMAP was solely used as a dimensionality reduction technique for visualization of high-dimensional data).

33. I do not understand well the rationale to perform the gene expression analysis first on one unique patient, and then see how it replicate in other patients through SCENIC (what about if integrating TAL1 patients via their expression?), to then see how it replicate in other type of ALLs. My main concern here is that the authors may be overfitting to a stem-like phenotype appearing in that particular patient (or the TAL1 subtype) and uncovering signatures that are more related to TAL1 T-ALL than to T-ALL.

Our response: To clarify, our initial analysis focused on identifying mechanisms associated with relapse, which led us to discover biological features related to stemness in the P2 patient. Specifically, we observed that numerous cells were in the G1 phase, exhibited a more immature phenotype, displayed a low metabolic state, and were enriched in pathways associated with treatment resistance. Notably, these stem-like characteristics comprised only a small population at the initial phase but increased significantly during relapse.

Following this exploration, we expanded our analysis to include all TAL1 patients, ultimately identifying that these features were not unique to the P2 patient but, importantly, common across the TAL1 subtype. We later confirmed that similar characteristics were also present in other subgroups of T-ALL, albeit at a lower frequency than for the TAL1 subtype. Importantly, the stemness score was calculated based on the stem-like cells identified in all TAL1 patients, ensuring that our findings are not limited to a single patient.

While we did perform integration of patients for our analysis initially, we chose to use pySCENIC for dimensional reduction in the TAL1 patients for several reasons. Unlike many other approaches, pySCENIC does not require batch correction because it relies on the inference of gene regulatory networks rather than direct expression values, making it inherently robust to batch effects. Batch correction methods, while useful for harmonizing datasets, can sometimes introduce artifacts or smooth out key biological differences, potentially obscuring meaningful signals.

By integrating all samples from the beginning, we arrive at the same conclusions regarding the stem-like phenotype and its associated signatures, even if detected signals in the batch-corrected analysis are less pronounced compared to the results obtained using pySCENIC (**Figure 1**). Further, the previously observed differences in the other cell populations are also diminished upon integration. By utilizing pySCENIC, we maintained the integrity of the biological signals, allowing us to identify distinct cell populations and their characteristics more effectively. The sequential approach allowed us to gain insights from a single patient before looking

at broader patterns, reflecting the natural progression of our scientific inquiry in the lab.

Regarding the concern about overfitting, we recognize that analyzing a single patient can lead to specific findings that may not generalize. We emphasize that our generalized analysis of all TAL1 patients was conducted independently, it was not informed by and did not "learn from" the prior analysis of a single patient. To start with a single patient was simply our chosen means of introducing the concept of stem-like cells expanding during the development of relapse to the reader, and also reflects the way this research study was pursued by the first author. Therefore, concerns about overfitting do, in our view, not apply. The consistent patterns observed across multiple patients strongly suggest that these features reflect broader biological characteristics of T-ALL rather than individual-specific variations.

Finally, we would like to emphasise that our revised manuscript provides novel independent evidence for generalisability of our findings: We now show that a stem-like molecular signature directly derived from our data is prognostically relevant for treatment induction failure in an independent large cohort of >1300 T-ALL patients (see our response to comment #36 further below). This important finding is now also highlighted in the abstract of our manuscript.

Figure 1: Integrated analysis of patients . a), b), c), e), g): UMAP visualization of all patients after anchor-based CCA integrations. a) distribution of patients. b) distribution of seurat clusters. Black arrow points towards the cluster with the highest stemness score. c) Enrichment of stemness score in integrated dataset. e) distribution of cells in different cell cycling phases. g) distribution of predicted cell types after mapping cells onto a human thymic reference dataset. d) Violinplot showing enrichment of stemness score among seurat clusters. Red dashed line reflects the threshold used for the definition of stem-like cells (0.25). f), h): Stacked barplots comparing distribution in different seurat clusters. e) distribution of cells in different cell cycling phases. h) distribution of predicted cell types.

34. It is not very clear how the different analyses were performed. While the Methods section and the Supp. Method section shed some light into them, there is no link to the scripts, and hence I cannot assess how the analyses were actually performed. On

these lines, were the samples from pre and post relapse in P2 integrated in the first section? If not, how do you know that the seen differences are not due to a batch effect?

Our response: We thank the reviewer for this comment. We now provide all relevant code for our analyses to enhance transparency and reproducibility within our manuscript (<https://github.com/Zaffe24/Costea-et-al.-2024>; https://github.com/Zaffe24/AS_VASAseq_sc_pipeline). To clarify, we did not perform any integration of the samples from pre- and post-relapse in P2. However, we can confidently rule out the possibility that the observed differences are due to technical batch effects. Each sample was analyzed in replicates—specifically, two plates for P2 initial (EMB-JL-v045 and EMB-JL-v046) and two plates for P2 relapse (EMB-JL-v053 and EMB-JL-v054)—and our analysis did not reveal any batch effects between these replicates (Suppl, Fig 1c and Suppl Fig 5d of the revised manuscript/Figure 2 of this document). We have included a visual representation for all patients in the supplementary figure of the revised manuscript.

Figure 2: UMAP representation of P2 initial and relapse technical replicates. Each sample was sorted in two 384-well plates for VASA-seq: two plates of initial sample (EMB-JL-v045 and EMB-JL-v046) and two plates of relapse sample (EMB-JL-v053 and EMB-JL-v054).

35. Also, it is not clear how genes were ranked/selected for the GSEA. Were ranked by expression? or ranked by LFC between stem-like and blasts? This is important to clarify further.

Our response: The list of DE genes between the stem-like cluster and the blasts was computed via the Seurat function FindMarkers() which applies the non-parametric Wilcoxon test. Only genes with a log₂ fold change (log₂FC) > 0.5 and an adjusted p-value < 0.05 (calculated using the Bonferroni correction method) were retained. To perform GSEA, we ran the prerank() module from the gseapy package on the list of DE genes ranked by Log₂FC. The module calculates an enrichment score for each gene set, assessing the degree to which members of the gene set are overrepresented at the top or bottom of the ranked list. The enrichment score is then normalized (NES) based on the gene set enrichment scores for all dataset permutations, which is the primary statistic used to compare enrichment results

across gene sets. We added these clarifications to the revised supplementary methods.

36. When computing the stemness score, the authors should compare their analysis to using gene expression directly. Again, I am not sure why authors did not perform traditional gene expression analysis.

Our response: We thank the reviewer for this insightful question, allowing us to make important additional observations, as we describe in the following:

First of all – to clarify – we did perform (classical) differential gene expression (DGE) analysis in order to define the genes that were used to compute the “stemness score”. We have explained our approach in the Methods section of the manuscript. Please find below a detailed explanation on how we calculated the stemness score and why we believe it is advantageous compared to traditional DGE analysis. To address the reviewer’s concern, we performed an independent analysis based on the top10 marker of the DGE analysis to validate our stemness score-based results. Please find a detailed explanation on both approaches below.

Initially, we identified the stem-like cell subpopulation shared in the UMAP across TAL1 patients using transcription factor (TF) activities derived from the pySCENIC analysis. Next, we sought to investigate whether stem-like cells can be also found in other T-ALL subgroups, which is why we expanded our analysis (now including TAL1, NKX2, TLX1 and HOXA subgroups). However, we could not observe a shared cell population in the UMAP among these subgroups as for the TAL1 cohort (which is in line with prior literature showing that each subgroup is driven by a distinct set of active TFs (Ferrando et al, Cancer Cell 2002; Liu et al Nature Genetics 2017, reviewed in Gianni et al, Cold Spring Harbor Perspectives in Medicine 2019)).

Therefore, in order to identify potential stem-like cells in other T-ALL subgroups, we instead constructed a “stemness score”. Specifically, we used the FindMarkers() function from the Seurat package which applies the non-parametric Wilcoxon test to identify genes that are significantly enriched in the stem-like cell population compared to other leukemic cell populations in the TAL1 cohort. The criteria for selection included a log₂ fold change (log₂FC) > 0.5 and an adjusted p-value < 0.05 (calculated using the Bonferroni correction method). This led to the identification of 601 differentially expressed (DE) genes.

These DE genes were next treated as a set of markers, which we used to calculate a stemness score for all individual cells – now including cells from other subgroups as well – in further analyses. For this, we applied the AddModuleScore() function in Seurat. This function calculates an aggregated expression score for a predefined set of genes – the target gene set – in our case, the DE genes from our initial analysis. Specifically, it computes the average expression of the target gene set per cell, comparing it to randomly selected control genes that have a similar expression distribution to the target gene set. The resulting score reflects the relative expression of the selected DE genes within each cell, which we used to infer stem-like characteristics. This way, we detected cells from other T-ALL subgroups to have a

similar stemness score as the stem-like cells of the TAL1 cohort (however, the overall frequency of stem-like cells was lower in the other T-ALL subgroups compared to the TAL1 cohort).

To address the reviewer's concern we have now also used the top10 marker genes of the TAL1 stem-like cohort (which we derived from the DGE analysis) and compared the expression in single cells of all subgroups (stem-like vs blasts). All 10 markers are significantly more enriched in the stem-like cohort than in the other leukemic cells (Suppl. Fig3 c of the revised manuscript/Figure 3 of this document). We have included this analysis in the supplemental material of our revised manuscript. However, since the stemness score integrates the expression of all 601 DE genes and calculates the average expression per cell, we see this approach as advantageous.

Figure 3: Enrichment of TAL1 patients' top 10 stemness markers in cells of all T-ALL subgroups. Normalized expression levels of top 10 stemness markers in blasts (grey, stemness score <0.25) and stem-like cells (blue, stemness score > 0.25). **** = $p < 2.22e^{-16}$ (two-sided t-test).

Second, additional analyses we pursued during paper revision indicate that the developed stemness score has the potential to be used as a prediction for clinical outcomes. In particular, we applied our stemness score (using the 601 DE genes) to 1,300 diagnostic T-ALL bulk transcriptomes from a recent publication (Pölönen et al, Nature, 2024, PMID: 39143224; published in August 2024). Our score, importantly, demonstrates a significant enrichment of stemness in patients with induction failure (25% of blasts after initial treatment for 29 days), indicating that our stemness score could serve as a useful prediction tool in the clinics (Fig. 6f of revised manuscript).

We also emphasise in this regard that analogous scores have been widely used in the past and present, with one of the first published examples being the LSC17 score in AML published 2016 in Nature (Ng et al, Nature 2016, PMID: 27926740). More recent examples using single cell RNA seq data include the generation of an inflammation score in AML (Lasry et al, Nature Cancer 2022, PMID: 36581735), a score associated with highly resistant cells in multiple myeloma (Cohen et al, Nature Medicine 2021; PMID: 33619369), and yet another example for a stemness score that was used to predict prognosis in APL (Jin et al, 2024, PMID: 38365836).

In summary, we apologize for any lack of clarity, and have ensured that the revised manuscript provides a clear explanation on the calculation of the stemness score, includes a comparison with the traditional DGE approach, and features an application of this stemness score in a separate cohort showing ability to predict induction failure in patients (see Fig. 6f of the revised the manuscript, also shown as Figure 4 in this document).

Figure 4: Comparison of average stemness score among patients with different treatment outcomes: Analysis of a public bulk RNA seq dataset of pediatric T-ALL patients at initial diagnosis (Pölonen et al, Nature 2024). Patients are grouped based on their morphological treatment response on day 29: M1 (<5% lymphoblasts), M2 (5 – 25% lymphoblasts) and M3 (>25% lymphoblasts). Calculation of stemness score per patient based on the average expression of previously identified 601 stemness markers.* = $p < 0.05$. *** = $p < 0.001$. Significance was analyzed using a two-sided t-test.

37. Also, the authors should consider using SCENIC+ (or even DeepSCENIC) which are updated versions of SCENIC (which is from 2017).

Our response: We clarify, while SCENIC+ is indeed a more recent version of the SCENIC pipeline, we purposely chose not to use it because it is specifically designed for integrating multi-omics data, particularly chromatin accessibility (e.g., scATAC-seq), into gene regulatory network analysis. According to the developers, SCENIC+ is not intended to replace SCENIC (i.e. pySCENIC) for analyses based solely on scRNA-seq data (<https://github.com/aertslab/scenicplus/discussions/177>), in line with our methodological choice.

Additionally, we clarify that we did not use the original SCENIC method published in 2017 but rather the pySCENIC implementation published in 2020, which was released later and offers several improvements, including better scalability and integration with scRNA-seq workflows. pySCENIC has become widely adopted in the field, offering a well-validated and effective approach for inferring gene regulatory networks. In fact, its usage has recently increased, including publications in prominent journals like *Nature Communications* and *Nature* (<https://citations.springernature.com/item?doi=10.1038/s41596-020-0336-2>). For instance, a 2024 study in *Nature* utilized pySCENIC to define the regulatory network changes in CD8 T cells, revealing how transcriptional shifts occur following asymmetric cell division (Lee et al, *Nature* 2024; PMID: 39198645). This illustrates, in our view, the ongoing relevance and robustness of pySCENIC in uncovering critical regulatory networks across various biological contexts. We made sure in our revised manuscript that the reasoning for our methodological choices is clear.

38. Regarding the alternative splicing section, I think this section adds a big value to the study and it is differential to what has been done before. However, the flaws in the computational identification of the stem-like cells downgrades a bit relevance of the section.

Our response: We much appreciate the positive feedback on the alternative splicing section! We made further revisions to the alternative splicing analyses in our revised manuscript, as detailed in our responses to comments #22 and #23. With respect to our clarification on how the stem-like cells were identified, we refer this reviewer to our responses to comments #33, #34 and #36 above).

In summary, I believe that the analyses performed to identify the stem-like population are confusing, the computational analysis is not well presented, and the mix of gene expression analyses with SCENIC analyses across the different cohorts looks like cherry-picking. These precludes me to fully believe that the identified population is indeed a stem-like population associated to T-ALL, rather than a specific population that *looks like* stem-cells that are found in specific patients.

Our response: We thank the reviewer for their comment. As detailed above in our responses to comments #33-#37, we carefully reviewed this part of our analysis to ensure its robustness in identifying and characterizing stem-like cells. With respect to the molecular origin of the stem-like cells uncovered in our study, we do stress that we do not claim that this cell population actually derives from stem cells – instead it is most likely that these cells derive from a more immature population, i.e. double negative T-cell progenitors. We see the

use of the term stem-like in this regard as appropriate, as this allows us to distinguish this cell population from actual stem cells. We further would like to highlight our novel findings indicating the utility of our “stemness score” for predicting treatment resistance, as revealed by analyses in a cohort of 1,300 primary T-ALL bulk RNA samples during manuscript revision.

To fully address this reviewer comment, we have also carefully revised our methods section ensuring adequate representation of our computational approaches used. Additionally, we made the code for our analysis openly available in the revised manuscript to enhance transparency and reproducibility(<https://github.com/Zaffe24/Costea-et-al.-2024>; https://github.com/Zaffe24/AS_VASAseq_sc_pipeline). Thank you again for your thoughtful comments, which were extremely helpful for further improving our manuscript.

Reviewer #3, expertise in stem cells, hematological cancers and alternative splicing (Remarks to the Author):

The manuscript by Costea et al reports that majority of pediatric T-ALL patients exhibit stem-like subpopulation, which although is a small percentage at initial diagnosis, expands significantly after relapses and is resistance to therapy. The authors performed substantial scRNA-sequencing (VASA-seq) to carefully characterize the gene expression, splicing pattern, and TFs. The main take home message is that resistant stem-like cells are resistant to treatment thus persist and expand; and these cells can be found in all molecular subtypes of T-ALL tested (although mostly in TAL1+ samples). Since the leukemia stem cell population is not well-defined in T-ALL, this study provides important insights into the specific regulon program and the molecular properties of T-ALL stem-like cells. Overall, the experimental data are solid and compelling. However, important functional experiments are missing to support the main conclusion.

Our response: We do appreciate the positive feedback regarding our work. At the same time, we have taken the reviewer’s comments into careful consideration, and in our revised manuscript, have incorporated functional experiments to further enhance the robustness of our study.

Major Points

39. What are the treatment histories of the relapsed patients used for scRNA-seq? Depending on the treatment, different “stem-like” cells may emerge based on genetic or epigenetic rewiring. It will be interesting to compare the stemness score and regulon patterns based on different treatment.

Our response: In our analyses, we observed that the T-ALL subtype (TAL1) and type-2 relapses are associated with a higher frequency of stem-like cells and corresponding regulon patterns. We agree that it would be interesting to explore in the future whether the emergence of stem-like cells might also be influenced by the treatment protocols used. However, the majority of the relapsed patients (8/13) were enrolled in the same study (ALL BFM 2000), while the remaining 5 patients were enrolled in 3 more studies (2/13 patients in ALL BFM 2009, 2/13 patients in COALL-3, 1/13 patients in COALL-9). Therefore, it is not feasible to robustly assess

treatment-specific effects within this cohort at this time but we have now stated in our discussion that future research studies should assess the influence of different treatment regimens.

40. The experimental approach in Fig. 6a-b is problematic. First, the patient selection for *in vitro* cytarabine treatment is peculiar. Why treat relapsed patients to test if stem-like cells expand since they already expanded? Samples at initial diagnosis prior to relapse will serve much better choice. Second, can authors explain why they choose patient p1 and p6? They don't have the highest stemness score (Fig. 3f). Lastly, for the *in vivo* treatment of p41 (Fig. 6e-h), it is important to show the expansion of stem-like cells are not simply the result of selective advantages of engrafting capacity. What's the cell composition in PDX without any treatment?

Our response: We appreciate the reviewer's comments. In response to these concerns on patient selection, we revised our *in-vivo* approach by focusing on initial diagnosis samples (P1 and P10) instead of relapse samples. These samples were selected because they had the highest fraction of stem-like cells at initial diagnosis and were part of the relapsing cohort. For *in-vivo* analyses, we evaluated both single-agent cytarabine treatment and a combinatorial regimen (doxorubicin, dexamethasone, vincristine) in our PDX model. To facilitate the comparability among the two samples (accounting for reviewer comment #28), we decided to compare the difference of stemness score following treatment instead of the patient-specific cluster changes. Our *in-vivo* data, notably, show that the stemness score of both samples was significantly enriched following treatment (Fig. 6d-e of the revised manuscript).

To further align our *in-vitro* and *in-vivo* data (accounting for reviewer comment #28), we also performed an additional *in-vitro* experiment using the P10 PDX sample (Fig 6d of the revised manuscript). As before, this new *in-vitro* experiment was conducted using relapsed disease samples, which allowed us to compare the results to previous *in-vitro* data of P1 and P6. As for the *in-vivo* data, we analyzed the stemness score following treatment rather than patient-specific cluster changes. The new *in-vitro* treated P10 sample showed a clear enrichment of stemness, similar to previous P1 and P6 samples.

Given the clear enrichment of stem-like cells observed in our *in-vivo* model, we decided not to further expand the *in-vitro* experiments to also include initial disease samples. Instead, we prioritized the *in-vivo* findings, as they provide a more physiologically relevant context and allowed us to optimize resource use.

Additionally, we acknowledge the concern regarding the potential selective advantage of certain cell populations influencing clonal composition. To address this, we engrafted PDX samples in mice without any treatment as a control (their leukemia cells were collected on the same day as those from the treatment groups), which ensures that the observed expansion of stem-like cells is not merely a result of differential engraftment capacity (see control group Fig 6d-e of revised manuscript).

41. Can the UMAPs and boxplots comparing diagnosis and relapse shown for all relapsed patients rather than only the 8 samples with >5% stem-like cells? Since

type I relapse (p11, p5, p9, and p4) generally has less stem-like cells, it will be interesting to see which subpopulation expands after treatment in these patients.

Our response: We agree that exploring alternative relapse mechanisms in patients with fewer stem-like cells is of interest. While our analyses have indicated that these mechanisms may be more patient-specific, we have now included the UMAPs and boxplots for all relapsed patients, as suggested by this reviewer (Suppl. Fig 5a-c of the revised manuscript), to provide a broader view and better understand the subpopulations expanding after treatment in these cases.

Minor Points

42. The panels g and f in Fig. 3 are flipped.

Our response: We have corrected the labelling in the revised manuscript.

43. We suggested to move Supplemental Fig. 2d to main Fig 3, which will help to interpret the data with pt #.

Our response: Thank you for the suggestion. We have moved this plot to the main Fig 3 as suggested.

44. How is the “stemness score” defined? Is it based on the regulon (TF) expression? This concept is frequently used to identify the stem-like cells (or stem-like) throughout the text. However, the exact method used to calculate the score cannot be found in supplemental method as mentioned by the authors.

Our response: We apologize if this has not been clearly stated, and we noticed that in the supplementary methods we used the term “stemness signature” instead of stemness score which might have led to confusion. We have corrected this in the revised manuscript. Please see our response to comment #36 to find a detailed explanation on the calculation of the stemness score.

45. Please provide a patient characteristic table.

Our response: Please see our response to comment #5 for our reply to this reviewer's question (i.e. such table is now provided in the supplement of our revised manuscript).

REVIEWERS' COMMENTS

Reviewer #2 (Remarks to the Author):

I thank the authors for the effort in responding all my concerns. I especially appreciate the independent validation on a large cohort, which make the results significantly more robust. I also highly appreciate the availability of the code, and how well organized and commented is.

Regarding my comments on using P2 first and then seeing finding replication on the rest of TAL1+ T-ALL patients, and, in lesser extend, in T-ALL patients has been addressed. Although, as a last request, I would like to see the patient distribution across clusters in the gene-expression-integrated clustering shown in Figure 1 of the response to author letter. In this regard, I would include that Figure as supp. figure to show that "orthogonal" analysis on the data is also able to identify the stemness population.

Finally, I understand the argument about the potential "oversmoothing" of gene-expression integration methods, and the capabilities of GRN-driven models to "overcome" batch effects. While not requested for this work, I wanted to point out to the authors recent population-level integration models that may be of use for the future: <https://www.nature.com/articles/s41592-023-02035-2>; <https://www.biorxiv.org/content/10.1101/2022.10.04.510898v2>; <https://www.nature.com/articles/s41587-023-01940-3>

Reviewer #2 (Remarks on code availability):

The code is very well organized and well commented.

Reply to reviewer #2:

We sincerely thank the reviewer for the thoughtful and constructive feedback, as well as the kind words regarding our validation efforts and the organization of the code.

As requested, we have now included the patient distribution across clusters from the gene-expression-integrated clustering (previously shown in Figure 1 of the response letter) as Supplementary Figure 3a-c in the revised manuscript. This addition highlights the ability of an orthogonal, gene expression-based approach to recover the stemness-enriched population, supporting the robustness of our findings.

We also appreciate the valuable references to recent population-level integration methods. While outside the scope of this study, we will certainly consider these approaches in future work.

Thank you again for your careful review and helpful suggestions.

Reviewer #3 (Remarks to the Author):

The manuscript has improved significantly. In particular, the new in vivo data (Fig. 6) has proved that stem-like population arises during drug treatment and is more resistance. Moreover, the application of the "stemness score" in a large cohort of T-ALL patients further validated these findings. Lastly, the method and supplemental section is more complete and much easier to follow in the revised version.

Reply to reviewer #3:

We thank the reviewer for the positive and encouraging feedback. We are pleased that the additional in vivo data and the validation of the stemness score in a large patient cohort strengthened the manuscript. We also appreciate your recognition of the improvements to the methods and supplemental sections. Your comments have been very helpful in refining and clarifying our work.

Reviewer #4 (Remarks to the Author):

Costea et al. investigate the mechanisms underlying T-ALL relapse through single-cell RNA sequencing on 13 matched pediatric T-ALL PDX samples at diagnosis and relapse, along with five non-relapsing PDX samples collected at diagnosis. Interestingly, 11 of 18 “patients” exhibited a subpopulation of T-ALL cells with a “stem-like” expression signature and splicing pattern. This subpopulation expanded substantially at relapse, indicating resistance to therapy. Increased “stemness” at diagnosis was associated with a higher risk of relapse. Chemotherapy resistance was validated through in vitro and in vivo drug testing.

This is a very interesting study. It is well written and significantly improved in its extensively revised version. The authors also provided a well-thought-out rebuttal letter to address the Reviewers’ comments on the previous submission.

Since the T-ALL stem cell population is not well-defined, this study provides potentially useful information by defining a specific regulon program/transcriptomic signature of T-ALL “stem-like” cells. Overall, the experimental data are solid and convincing. The revised version somewhat improved the functional validation, which was the main weak point of the original submission.

One problem with the present submission is that, throughout the manuscript, PDX are referred to as “patients”. This is misleading and should be corrected. Although the Authors state that in a previous study (surprisingly) they did not observe clonal selection of genetic/ epigenetic differences between patients and PDX, primary samples ex vivo from patients are not the same as PDX.

Reply to reviewer #4:

We thank the reviewer for the thoughtful and supportive comments, and we are pleased that the revised version was found to be significantly improved, particularly in terms of functional validation.

We fully agree with the reviewer’s important point regarding the terminology used to describe the PDX models. To avoid confusion and ensure clarity, we have carefully revised the manuscript to consistently refer to these samples as PDXs rather than “patients.” This change has been applied throughout the text, including the figures, legends, and supplemental materials.

We appreciate your valuable feedback, which helped us further improve the accuracy and clarity of our work.

Reviewer #5 (Remarks to the Author):

Reply to reviewer #5:

Thank you for your contribution to the review process. We appreciate your involvement and the support for Early Career Researchers through this initiative.